# Screening for consistency and contamination within and between bottles of 29 herbal supplements

**Maren E. Veatch-Blohm**[ID]*, **Iris Chicas, Kathryn Margolis, Rachael Vanderminden, Marisa Gochie, Khusmanie Lila**

Department of Biology, Loyola University Maryland, Baltimore, Maryland, United States of America

* mblohm@loyola.edu

**Data Availability Statement:** All data files are available from the Harvard Dataverse database (https://doi.org/10.7910/DVN/TRLTI3).

## Abstract

In the United States the marketing of dietary supplements, of which the majority are herbal supplements, is currently a multibillion-dollar industry involving use from over half of the adult population. Due to their frequency of use and the lack of regulation of herbal supplements by the Food and Drug Administration (FDA) it is important for the health and safety of consumers to know about consistency of supplements and any possible contamination by harmful products, such as heavy metals or microorganisms. The purpose of the study was to determine consistency and contamination within and between bottles of common herbal supplements. Duplicate bottles of 29 herbal supplements were tested for consistency for antioxidant activity, phenolic concentration and flavonoid concentration under methanolic and water extraction. The supplements were also analyzed for the presence of metals and fungal contaminants. For all of the supplements tested there was high variability around the mean in antioxidant activity, phenolic concentrations and flavonoid concentrations, with coefficients of variation (CV) ranging from 0–120. Zinc was found in almost 90% of the supplements, nickel in about half of the supplements and lead in none of the supplements. Approximately 60% of the supplements contained fungal isolates. Although the majority of the fungi that were found in the supplements are generally not hazardous to human health, many of them could be problematic to sensitive groups, such as immunocompromised individuals. The data, which demonstrates contamination and a lack of consistency, in conjunction with previous studies on supplement contamination, strengthen the case that the FDA should regulate over-the-counter herbal supplements the same way that they regulate food and drugs. Until such time it is crucial that consumers are informed that many of the supplements that they take may lack the standardization that would reduce the chance of contamination and lead to consistency from one pill to the next.

## Introduction

Herbal remedies have existed for thousands of years and the demand for herbal supplements worldwide has increased significantly over the past few decades. Every year more consumers use over-the-counter herbal supplements for needs varying from digestive health and

**Funding:** The authors received no specific funding for this work.

**Competing interests:** The authors have declared that no competing interests exist.

improved energy to relief from depression. Herbal medicines are thought to enhance and restore normal physiological functions by facilitating the body's innate self-healing capabilities [1]. Additionally, many marketed herbal products are mixtures or parts of organic chemicals that come from any raw or processed part of a plant; because these products come from plants consumers assume that they are healthy [2]. The World Health Organization estimates that about 80% of the populations in Asian and African countries take herbal medicine as their only form of medication and treatment [3], while in Europe about 19% of adults use 'Plant Food Supplements' [4]. Dietary supplements are used by 50 to 70% of adults in the United States [5, 6] of which herbal supplements are the most popular subset [7]. In 2018, the sales of herbal supplements in the U.S. went up almost 9.4%, which represented the thirteenth straight year of sales increases [8]. Overall net profits from the sale of herbal supplements within the U. S. are over eight billion dollars per year [8]. Over the past ten years herbal supplements that are marketed as adaptogens, to enhance immune function, reduce stress and anxiety have consistently been among the top selling supplements accounting for a high percent of the sales in the supplement market [8, 9].

A 2015 report released by the New York state's attorney general revealed that store brand supplements did not contain the ingredients listed on the tested product labels up to 80% of the time [10]. Newmaster et al. [11] investigated 44 over-the-counter supplements utilizing DNA barcoding and found that the ingredients listed on the tested product labels were not found in the products. Adulteration using DNA barcoding has been found worldwide with a range of adulteration from 23–79% of tested supplements [12, 13]. In addition, in July 2019, Amazon sent a notice to its customers warning that some nutritional supplements recently sold on their site may have been fake [14]. Unlike pharmaceutical drugs, herbal supplements are not regulated by the Food and Drug Administration (FDA) and can only be recalled when there is substantive proof of harm or when contaminants are present that would change the regulatory status of the supplement [15]. Most Americans consuming over-the-counter herbal supplements often do not realize that the FDA does not regulate herbal supplements the same way that they regulate food and drugs [16].

The lack of regulation of herbal supplements by the FDA only increases the susceptibility of these products to contamination [17] with no assurance of purity, integrity or efficacy. Contamination may be due to growth, processing or storage conditions or may be due to the intentional addition of unlisted ingredients. Geyer et al. [18] studied over 600 nutritional supplements between 2001 and 2002 and found that 15% of the supplements purchased in 13 different countries were contaminated with anabolic steroids. In another study, Chinese herbal medicines (produced and sold in China) also demonstrated contamination with heavy metals and microbes [19], which may come from cultural conditions or production processes. Fungal contamination, particularly fungi that produce mycotoxins, many of which are known carcinogens or neurotoxins, is often associated with storage conditions [20–22]. Contamination may also occur through the accidental or intentional inclusion of other plant species [23–26]. For example, different species of *Echinacea* are often substituted for each other; even though they do not all have the same efficacy [27]. In 1997, two previously healthy individuals experienced irregular heartbeats after using plantain supplements contaminated with foxglove (*Digitalis* sp.), which contains cardiac glycosides [24]. In addition, contamination with allergens could lead to severe allergic reactions in sensitive groups. Perhaps the most troubling source of contamination is the addition of pharmaceutical grade drugs to herbal supplements, which could lead to adverse health effects [17, 28–31].

It may also be difficult, in field collected materials, to completely eliminate microbial and mycotoxin contaminants from the area where the plants are grown [32, 33]. Field collections pose other risks as well: plants could naturally take up metal contaminants from the soil and

water they are exposed to during growth [32]. Although there have been many studies examining purity and effectiveness of herbal supplements, with hundreds of different supplements from hundreds of suppliers it is critical to evaluate as many supplements as possible. The purpose of the research was to determine the degree of both consistency and contamination within duplicate bottles of 29 herbal supplement products, totaling 58 bottles. The majority of the supplements that were chosen for this analysis were among the most commonly sold on a year-to-year basis in the United States, which typically fall in the categories of general adaptogens, stress and depression relief, and immune response [8, 9].

## Materials and methods

### ASupplement selection

The supplement manufacturers were Nature's Way, Spring Valley, NOW, and Sundown Naturals. Two bottles each from a single supplier were tested for the following supplements: Aloe, Biotin, Cranberry, Raspberry, Reishi, Silent Night, Stress Formula, and Yarrow. Two bottles each from two different suppliers were tested for the following supplements: Astragalus, Echinacea, Echinacea Goldenseal, Ginger, Ginseng, and Rhodiola. All Spring Valley supplements were purchased in person at Walmart in Cockeysville, Maryland as were the Sundown Naturals Turmeric and Stress Formula. All other supplements were purchased online through the Amazon Marketplace. Two bottles each from three different suppliers were tested for the following supplements: St. John's Wort, Turmeric and Valerian Root. All bottles were sealed and stored away from light at room temperature (between 20 and 25°C) until initial analysis. After opening, the bottles were stored with the lids tightly closed under the same temperature and light conditions. The humidity within the lab ranged between 55 and 65%. In addition, each bottle was stored with the included silica pack to reduce moisture accumulation within the bottles. Additional information about each supplement, including lot number and expiration date, is found in Table 1.

### Sample extraction

Ten capsules were removed at random from each bottle and from each capsule sub-samples were weighed out for a variety of extraction methods (methanol, hot water and nitric acid). Four subsamples were taken from each capsule. Two subsamples were taken to analyze the methanolic and water soluble fractions of each supplement and extracted according to the method of Cai et al. [34] with the following modifications such that the assays were conducted on the same day as the extraction. One subsample was extracted with 90% methanol and one with boiling water. After methanol or water addition, the samples were vortexed for 10 seconds and put in a shaker at 150 rpm at 25°C for 60 minutes. The samples were then centrifuged for 5 minutes at 5,000 g followed by removal of the supernatant. Immediately after supernatant removal, three assays were conducted on each fraction. All of the extractions and tests, with the exception of the supplement stability tests were conducted from October 2016 through April 2017. The supplement stability tests were conducted in January and February 2019.

### Antioxidant, phenolic and flavonoid assays

Antioxidant capacity, expressed in Trolox equivalents (TE), was tested with the Ferric Reducing Antioxidant Power (FRAP) assay [35]. This assay utilized a standard curve created with Trolox with concentrations ranging from 0–1500 $\mu$mol$\bullet$ L$^{-1}$ Total phenolic concentration, expressed in gallic acid equivalents (GAE), was tested using the Folin-Ciocalteu method, with a standard curve produced using a range of gallic acid concentration from 0 to 0.4 mg$\bullet$ ml$^{-1}$

**Table 1. List of supplements and suppliers for 58 bottles of over the counter herbal supplements.**

| Supplier* | Supplement | Bottle | Batch # | EXP Date | Bottles different* |
|---|---|---|---|---|---|
| Nature's Way | Aloe | 1 | 20043572 | Jun-18 | Yes |
| Nature's Way | Aloe | 2 | 20066063 | Oct-19 | 2/6 |
| Nature's Way | Astragalus | 1 | 20044498 | May-20 | Yes |
| Nature's Way | Astragalus | 2 | 20066304 | Oct-21 | 4/6 |
| NOW | Astragalus | 1 | 1881061 | May-18 | Yes |
| NOW | Astragalus | 2 | 1881061 | May-18 | 4/6 |
| NOW | Biotin | 1 | 1904240–2042 | Jan-18 | Yes |
| NOW | Biotin | 2 | 2110421–1022 | Jan-19 | 5/6 |
| Nature's Way | Cranberry | 1 | 20041000 | May-20 | Yes |
| Nature's Way | Cranberry | 2 | 20067173 | Oct-21 | 4/6 |
| Spring Valley | Echinacea | 1 | 800025 | Jul-18 | Yes |
| Spring Valley | Echinacea | 2 | 800025 | Jul-18 | 4/6 |
| Sundown Naturals | Echinacea | 1 | 988248–07 | Apr-20 | Yes |
| Sundown Naturals | Echinacea | 2 | 988248–07 | Apr-20 | 5/6 |
| Nature's Way | Echinacea Goldenseal | 1 | 20066301 | Nov-21 | Yes |
| Nature's Way | Echinacea Goldenseal | 2 | 20066301 | Nov-21 | 3/6 |
| Spring Valley | Echinacea Goldenseal | 1 | 800009 | Jun-18 | Yes |
| Spring Valley | Echinacea Goldenseal | 2 | T800006 | Jan-19 | 4/6 |
| NOW | Ginger | 1 | 2084551–0058 | Nov-18 | Yes |
| NOW | Ginger | 2 | 2084551–0121 | Nov-18 | 3/6 |
| Spring Valley | Ginger | 1 | 987324–02 | Mar-20 | Yes |
| Spring Valley | Ginger | 2 | 994218–01 | Sep-20 | 2/6 |
| Sundown Naturals | Ginseng Xtra | 1 | 888712–05 | Mar-17 | Yes |
| Sundown Naturals | Ginseng Xtra | 2 | missing | missing | 6/6 |
| Spring Valley | Korean Panax Ginseng | 1 | 1039222 | May-18 | Yes |
| Spring Valley | Korean Panax Ginseng | 2 | 1054619 | Feb-19 | 6/6 |
| Nature's Way | Red Raspberry Leaf | 1 | 20065651 | Sep-21 | Yes |
| Nature's Way | Red Raspberry Leaf | 2 | 20065651 | Sep-21 | 4/6 |
| Nature's Way | Reishi | 1 | 20039110 | Feb-18 | Yes |
| Nature's Way | Reishi | 2 | 20064691 | Sep-19 | 3/6 |
| Nature's Way | Rhodiola | 1 | 20066944 | Nov-18 | Yes |
| Nature's Way | Rhodiola | 2 | 20066944 | Nov-18 | 4/6 |
| NOW | Rhodiola | 1 | 2025719–0221 | Jun-21 | Yes |
| NOW | Rhodiola | 2 | 2105309–2246 | Dec-21 | 6/6 |
| Nature's Way | Silent Night | 1 | 20056474 | Feb-21 | Yes |
| Nature's Way | Silent Night | 2 | 20064590 | May-21 | 5/6 |
| Nature's Way | St. John's Wort | 1 | 20066202 | 0ct-19 | Yes |
| Nature's Way | St. John's Wort | 2 | 20066202 | 0ct-19 | 2/6 |
| Spring Valley | St. John's Wort | 1 | 1041145 | Jul-18 | Yes |
| Spring Valley | St. John's Wort | 2 | 1048622 | Aug-18 | 4/6 |
| Sundown Naturals | St. John's Wort | 1 | 992660–03 | Jun-19 | Yes |
| Sundown Naturals | St. John's Wort | 2 | 992660–03 | Jun-19 | 6/6 |
| Sundown Naturals | Stress formula | 1 | 983673–01 | Feb-19 | Yes |
| Sundown Naturals | Stress formula | 2 | 983245–03 | Mar-19 | 4/6 |
| Nature's Way | Turmeric | 2 | 20040681 | Apr-18 | Yes |
| Nature's Way | Turmeric | 1 | 20045321 | Jul-18 | 3/6 |
| Spring Valley | Turmeric | 1 | 988358–01 | Jun-19 | Yes |

(*Continued*)

**Table 1.** (Continued)

| Supplier* | Supplement | Bottle | Batch # | EXP Date | Bottles different* |
|---|---|---|---|---|---|
| Spring Valley | Turmeric | 2 | 28008 | Oct-19 | 3/6 |
| Sundown Naturals | Turmeric | 1 | 976191–02 | Dec-19 | Yes |
| Sundown Naturals | Turmeric | 2 | 976191–02 | Dec-19 | 2/6 |
| Nature's Way | Valerian Root | 1 | 20066814 | Oct-21 | Yes |
| Nature's Way | Valerian Root | 2 | 20066819 | Oct-21 | 4/6 |
| Spring Valley | Valerian Root | 1 | 1042635 | Aug-18 | Yes |
| Spring Valley | Valerian Root | 2 | 1048489 | Nov-18 | 5/6 |
| Sundown Naturals | Valerian Root | 1 | 987221–01 | May-19 | Yes |
| Sundown Naturals | Valerian Root | 2 | 987221–01 | May-19 | 3/6 |
| Nature's Way | Yarrow | 1 | 20051391 | Oct-20 | Yes |
| Nature's Way | Yarrow | 2 | 20065763 | Oct-21 | 3/6 |

*For each bottle of each supplement the batch number and expiration date are included. Also indicated are if the bottles had significant differences on any of the antioxidant, phenolic and flavonoid assays run and on how many of the six assays they had significant differences.

[36]. Flavonoid concentration, expressed in catechin equivalents (CE), was tested using the AlCl$_3$ precipitation method, with a standard curve created using a range of (+)-catechin hydrate from 0 to 1000 μg • g$^{-1}$ [37]. All assay values were calculated and standardized based on the weight of the sample that was extracted. To assess supplement stability, the three assays were conducted again on methanol and hot water extracts from 5 additional capsules per bottle of the Echinacea, Echinacea-Goldenseal, Turmeric, and Valerian Root two years after the initial testing. These supplements were chosen such that two supplements from two suppliers and two from three suppliers were tested.

## Metal extraction and gross physical contamination

Each bottle was also tested for the presence of metals, gross physical contamination and fungi. One subsample from each capsule (0.05 g) was wet digested with nitric acid for flame atomic absorption spectroscopy (FAAS) and prepped according to Pomper and Grusak [38] to measure levels of nickel, zinc, lead, copper, and chromium. To check for gross physical contamination of the samples, the remaining sub-sample from each capsule was screened under a light-dissecting microscope. The samples were scanned quickly for uniformity within a capsule and among capsules within the same bottle and between different bottles. In addition to the above tests, multiple capsules were pooled in order to assess fungal contamination in the supplements.

## Fungal isolation

This fungal screen was conducted twice for each bottle based on the method of Singh et al. [39]. For each extraction five grams were taken from each bottle and added to forty-five milliliters of sterile water. Samples were mixed well and allowed to settle for 24 hours. Supernatant was used to inoculate sterile petri dishes, in duplicate, containing Sabouraud Dextrose Agar. In total, four plates were inoculated per bottle. After inoculation, the petri dishes were monitored for fungal growth for five to seven days and contaminants were classified based on visual characteristics. In addition, eight control plates were inoculated with sterile water only under the same plating conditions.

## Statistical analysis

For each bottle of each supplement, the means and standard error were calculated for the water, methanol, and nitric acid extractions and are the only calculations done for metal content within the supplements. In addition, the coefficient of variation (CV) was calculated for the water and methanol extractions. Comparisons for the FRAP antioxidant capacity and phenolic and flavonoid contents among suppliers and bottles from the same supplier were done using the general linear model in the FitModel platform of JMP and comparison among bottles and supplements compared using LSD Student's T-Test ($\alpha$ level of 0.05) [40]. A runs test was also conducted to test for randomness of rankings among bottles, suppliers and supplements using Minitab® Statistical Software with an $\alpha$ level of 0.05 [41]. A significant runs test means suppliers and/or supplements clump together. To determine if there had been a change in the average values for the FRAP antioxidant, phenolic and flavonoid assays after two years of supplement storage the data was analyzed using a one-tailed t-test with an $\alpha$ level of 0.05 and a test value of $< -25\%$ change using Minitab® Statistical Software [41].

## Results

### Antioxidant, phenolic and flavonoid assays

There were significant differences not only among suppliers for individual supplements (Table 2), but between bottles from the same supplier (Table 3).

There were significant differences between bottles for at least two assays for each supplement (Tables 2 and 3). When there were 2 or more suppliers per supplement there were always differences among suppliers for at least two of the assays. Ten of the supplements that were tested had duplicate bottles that were from the same batch number (Table 1). The least variation between bottles was Ginger from Spring Valley and St. John's Wort from Nature's Way (only a significant difference in a third of the tests), while for Ginseng Xtra, Korean Panax Ginseng, St. John's Wort from Sundown Naturals the bottles were significantly different from each other in 100% of the tests. Three of the supplements (St. John's Wort, Turmeric and Valerian Root) came from the three main suppliers. Data from the methanolic extraction of these three supplements is found in (Fig 1a–1c). Similar differences between bottles and suppliers were obtained for the water extraction; therefore, only the data from the methanolic extractions is shown. Firstly, the figures illustrate the differences among supplements. Turmeric had the highest antioxidant capacity and flavonoid concentration, while valerian root was the lowest for all three assays. The data also illustrate differences among suppliers for the same supplement. For example, for both St. John's Wort and Turmeric there was almost always a supplier that was significantly lower for the tested compounds than the other suppliers; however, the supplier with the lowest values was not consistent across supplements. Finally, the data illustrate the differences that could occur between bottles from the same supplier. This was particularly evident in the Turmeric samples, where bottle 1 from Spring Valley had significantly lower antioxidant capacity (Fig 1a), phenolic concentration (Fig 1b) and flavonoid concentration (Fig 1c) than not only other suppliers, but also the other bottle from the same supplier. This bottle was also found to be much lighter in color than all other Turmeric bottles that were sampled.

The coefficient of variation (CV) was used to show the relative standard deviation among suppliers, bottles and supplements (Table 4), which is particularly valuable when the values involved can be different by orders of magnitude and demonstrate how much variability there were within each supplement. There is a wide range of CVs (S3 Table) for each supplement with the greatest CV found in Valerian Root, where the most consistent bottle had a CV for

**Table 2. The mean ± SE FRAP antioxidant capacity (A), phenolic content (P) and flavonoid content (F) from 58 bottles of over the counter herbal supplements extracted with hot water and methanol by supplier\*.**

| Supplier | Supplement | Water Extraction | | | Methanol Extraction | | |
|---|---|---|---|---|---|---|---|
| | | A (TE) | P (GAE) | F (CE) | A (TE) | P (GAE) | F (CE) |
| Nature's Way | Aloe | 76637 ± 4295^ | 8.94 ± 0.43^ | 4063 ± 359^ | 85256 ± 8460^ | 10.64 ± 0.52^ | 5646 ± 217^ |
| Nature's Way | Astragalus | 9342 ± 1294[B] | 1.91 ± 0.08[A] | 1308 ± 417[A] | 8794 ± 1827[A] | 1.09 ± 0.05[B] | 542 ± 59[B] |
| NOW | Astragalus | 22629 ± 1525[A] | 1.66 ± 0.16[A] | 675 ± 13[A] | 10114 ± 520[A] | 1.89 ± 0.18[A] | 725 ± 23[A] |
| NOW | Biotin | 412 ± 91^ | 0.23 ± 0.02^ | 555 ± 241^ | 4060 ± 915^ | 0.41 ± 0.04^ | 1169 ± 348^ |
| Nature's Way | Cranberry | 26359 ± 1781^ | 3.26 ± 0.26^ | 2185 ± 240^ | 25754 ± 1780^ | 1.55 ± 0.12^ | 3910 ± 903^ |
| Spring Valley | Echinacea | 200190 ± 31988[A] | 10.59 ± 0.60[A] | 15962 ± 566[A] | 11163 ± 170[A] | 1.04 ± 0.05 | 1793 ± 101[A] |
| Sundown Naturals | Echinacea | 94474 ± 15013[B] | 6.13 ± 0.61[B] | 7790 ± 714[B] | 8311 ± 401[B] | 1.34 ± 0.51 | 521 ± 84[B] |
| Nature's Way | Echinacea Goldenseal | 98618 ± 3687[A] | 3.97 ± 0.41[B] | 8206 ± 247[b] | 30641 ± 640[A] | 4.60 ± 0.72[A] | 3880 ± 140[A] |
| Spring Valley | Echinacea Goldenseal | 57687 ± 9677[B] | 6.74 ± 0.46[A] | 9652 ± 664[a] | 14733 ± 1783[B] | 1.57 ± 0.19[B] | 1308 ± 422[B] |
| NOW | Ginger | 88335 ± 2353[A] | 3.43 ± 0.15[A] | 1243 ± 36[B] | 373256 ± 46229[A] | 7.48 ± 0.14[A] | 6660 ± 1006[A] |
| Spring Valley | Ginger | 72285 ± 3836[B] | 2.98 ± 0.22[A] | 1444 ± 48[A] | 140170 ± 46229[B] | 5.09 ± 0.30[B] | 3472 ± 117[B] |
| Sundown Naturals | Ginseng Xtra | 5950 ± 869[A] | 0.76 ± 0.10[A] | 303 ± 68[B] | 3035 ± 401[B] | 0.59 ± 0.04[A] | 455 ± 96[A] |
| Spring Valley | Korean Panax Ginseng | 73 ± 23[B] | 0.18 ± 0.04[B] | 775 ± 217[A] | 4770 ± 1102[A] | 0.32 ± 0.03[B] | 26 ± 9[B] |
| Nature's Way | Red Raspberry Leaf | 488121 ± 34130^ | 12.38 ± 0.36^ | 8272 ± 130^ | 86122 ± 3019^ | 4.30 ± 0.18^ | 4081 ± 89^ |
| Nature's Way | Reishi | 15575 ± 1729^ | 2.29 ± 0.18^ | 1469 ± 194^ | 5536 ± 1153^ | 0.56 ± 0.04^ | 1768 ± 505^ |
| Nature's Way | Rhodiola | 853458 ± 28330[A] | 15.59 ± 0.52[A] | 15566 ± 315[A] | 757413 ± 109746[A] | 17.07 ± 0.88[A] | 18741 ± 1404[A] |
| NOW | Rhodiola | 602570 ± 46729[B] | 7.59 ± 1.88[B] | 7764 ± 739[B] | 222758 ± 42227[B] | 13.66 ± 0.96[B] | 10941 ± 852[B] |
| Nature's Way | Silent Night | 70378 ± 4516^ | 3.38 ± 0.36^ | 5324 ± 336^ | 12044 ± 899^ | 3.58 ± 0.22^ | 1515 ± 203^ |
| Nature's Way | St. John's Wort | 206984 ± 12697[A] | 8.00 ± 0.47[B] | 8755 ± 255[B] | 53124 ± 3352[C] | 4.14 ± 0.16[C] | 4278 ± 98[C] |
| Spring Valley | St. John's Wort | 180944 ± 21774[A] | 17.00 ± 0.68[A] | 16036 ± 714[A] | 93851 ± 15547[B] | 16.64 ± 0.45[A] | 10028 ± 760[B] |
| Sundown Naturals | St. John's Wort | 196151 ± 23313[A] | 17.42 ± 0.71[A] | 16816 ± 1013[A] | 154977 ± 12703[A] | 13.77 ± 0.49[B] | 11177 ± 301[A] |
| Sundown Naturals | Stress formula | 93095 ± 15299^ | 6.53 ± 0.34^ | 10698 ± 366^ | 18168 ± 1524^ | 1.23 ± 0.25^ | 1176 ± 164^ |
| Nature's Way | Turmeric | 1762 ± 84[C] | 0.68 ± 0.07[C] | 0 ± 0[B] | 183344 ± 10703[C] | 14.03 ± 0.71[B] | 339869 ± 21356[A] |
| Spring Valley | Turmeric | 5948 ± 693[B] | 1.03 ± 0.06[B] | 2984 ± 794[A] | 125930 ± 17877[B] | 8.56 ± 1.69[C] | 49705 ± 7829[C] |
| Sundown Naturals | Turmeric | 17353 ± 543[A] | 1.33 ± 0.16[A] | 1786 ± 208[A] | 266502 ± 18262[A] | 17.90 ± 1.11[A] | 176657 ± 7828[B] |
| Nature's Way | Valerian Root | 29662 ± 1460[B] | 2.77 ± 0.08[B] | 3106 ± 101[B] | 11684 ± 767[B] | 1.47 ± 0.06[B] | 1808 ± 76[A] |
| Spring Valley | Valerian Root | 10390 ± 3289[C] | 1.23 ± 0.20[C] | 2633 ± 416[C] | 6401 ± 381[C] | 0.79 ± 0.18[C] | 355 ± 59[B] |
| Sundown Naturals | Valerian Root | 43176 ± 2577[A] | 3.88 ± 0.24[A] | 3424 ± 99[A] | 15973 ± 772[A] | 1.81 ± 0.09[A] | 1854 ± 84[A] |
| Nature's Way | Yarrow | 141389 ± 10630^ | 7.67 ± 0.66^ | 11126 ± 665^ | 37482 ± 913^ | 2.72 ± 0.13^ | 3100 ± 300^ |

\* The assays include the Ferrric Reducing Antioxidant Power (FRAP) assay for antioxidant capacity expressed as Trolox equivalents per gram, the Folin-Ciocalteu method for phenolics expressed as gallic acid equivalents per gram (GAE), and the AlCl3 precipitation assay for flavonoids expressed as +-catechin equivalents per gram. The values are expressed as the mean ± SE (n = 20) for each extraction from each supplier of each supplement. Means within the same supplement with different capital letters are significantly different based on the LSD Student's T-test (± of 0.05). For specific p-values for each supplement supplier see S1 Table.

^—Means followed by this symbol indicate a supplement where there were no duplicate suppliers.

antioxidant activity that was 74 times lower than the bottle with the highest CV. This inconsistent pattern was observed throughout most of the bottles within the same supplier as well as different supplements within different suppliers. In addition, the rank runs test showed that for the bottles from the same supplier the bottles did not generally rank together (NS runs tests, Table 4).

## ASupplement stability

To get an indication of supplement stability, the antioxidant, phenolic and flavonoid assays on nine of the supplements (a total of 18 bottles) were conducted two years after the initial assays

**Table 3. The mean ± SE FRAP antioxidant capacity (A), phenolic content (P) and flavonoid content (F) from 58 bottles of over the counter herbal supplements extracted with hot water and methanol by bottle and supplier\*.**

| Supplier | Supplement | Bottle | Water Extraction | | | Methanol Extraction | | |
|---|---|---|---|---|---|---|---|---|
| | | | A (TE) | P (GAE) | F (CE) | A (TE) | P (GAE) | F (CE) |
| Nature's Way | Aloe | 1 | 66460 ± 7203[a] | 8.54 ± 0.86[a] | 4193 ± 735[a] | 496322 ± 1549[a] | 11.20 ± 0.80[a] | 5276 ± 315[a] |
| Nature's Way | Aloe | 2 | 86814 ± 1728[b] | 9.34 ± 0.13[a] | 3933 ± 33[a] | 120880 ±4217[b] | 10.08 ± 0.67[a] | 6016 ± 264[a] |
| Nature's Way | Astragalus | 1 | 13014 ± 2001[a] | 1.81 ± 0.14[a] | 2030 ± 786[a] | 16712 ± 364[a] | 1.21± 0.08[a] | 308 ± 43[b] |
| Nature's Way | Astragalus | 2 | 5670 ± 269[b] | 2.02 ± 0.09[a] | 587 ± 22[b] | 875 ± 185[b] | 0.97 ± 0.02[b] | 776 ± 28[a] |
| NOW | Astragalus | 1 | 28942 ± 857[a] | 2.05 ± 0.25[a] | 667 ± 15[a] | 11878 ± 477[a] | 1.16 ± 0.06[b] | 757 ± 22[a] |
| NOW | Astragalus | 2 | 16316 ± 482[b] | 1.26 ± 0.11[b] | 685 ± 21[a] | 8351 ± 475[b] | 2.62 ± 0.12[a] | 692 ± 40[a] |
| NOW | Biotin | 1 | 32 ± 9[b] | 0.22 ± 0.03[a] | 1111 ± 421[a] | 8044 ± 105[a] | 0.29 ± 0.03[b] | 2338 ± 456[a] |
| NOW | Biotin | 2 | 792 ± 56[a] | 0.24 ± 0.01[a] | 0 ± 0[b] | 76 ± 39[b] | 0.52 ± 0.04[a] | 0 ± 0[b] |
| Nature's Way | Cranberry | 1 | 24622 ± 3459[a] | 4.21 ± 0.29[a] | 2255 ± 487[a] | 19145 ± 1250[b] | 1.34 ± 0.17[b] | 589 ± 64[b] |
| Nature's Way | Cranberry | 2 | 28095 ± 869[a] | 2.31 ± 0.06[b] | 2114 ± 71[a] | 32362 ± 1455[a] | 1.77 ± 0.15[a] | 7230 ± 994[a] |
| Spring Valley | Echinacea | 1 | 330437 ± 19358[a] | 9.49 ± 1.00[a] | 14186 ± 663[b] | 11249 ± 231 | 1.01 ± 0.05[a] | 1421 ± 95[b] |
| Spring Valley | Echinacea | 2 | 69943 ± 13259[b] | 11.70 ± 0.52[a] | 17738 ± 460[a] | 11077 ± 258 | 1.07 ± 0.09[a] | 2164 ± 61[a] |
| Sundown Naturals | Echinacea | 1 | 29672 ± 1656[b] | 4.16 ± 0.33[b] | 5358 ± 405[b] | 9212 ± 565[a] | 0.94 ± 0.09[a] | 700 ± 136[a] |
| Sundown Naturals | Echinacea | 2 | 159277 ± 3976[a] | 8.10 ± 0.77[a] | 10222 ± 821[a] | 7409 ± 423[b] | 1.73 ± 1.03[a] | 341 ± 62[b] |
| Nature's Way | Echinacea Goldenseal | 1 | 98028 ± 7015[a] | 5.53 ± 0.39[a] | 7797 ± 395[a] | 32553 ± 929[a] | 2.36 ± 0.10[b] | 3956 ± 70[a] |
| Nature's Way | Echinacea Goldenseal | 2 | 99208 ± 2847[a] | 2.42 ± 0.07[b] | 8615 ± 252[a] | 28728 ± 230[b] | 6.84 ± 1.04[a] | 3804 ± 277[a] |
| Spring Valley | Echinacea Goldenseal | 1 | 85953 ± 14036[a] | 6.55 ± 0.73[a] | 10709 ± 797[a] | 21867 ± 1432[a] | 2.33 ± 0.13[a] | 2093 ± 722[a] |
| Spring Valley | Echinacea Goldenseal | 2 | 29421 ± 1134[b] | 6.94 ± 0.59[a] | 8477 ± 986[a] | 7599 ± 265[b] | 0.80 ± 0.05[b] | 436 ± 114[b] |
| NOW | Ginger | 1 | 81890 ± 2301[b] | 3.25 ± 0.13[a] | 1181 ± 35[a] | 559238 ± 35560[a] | 7.38 ± 0.17[a] | 3803 ± 207[b] |
| NOW | Ginger | 2 | 94782 ± 2976[a] | 3.60 ± 0.26[a] | 1306 ± 59[a] | 187273 ±8513[b] | 7.58 ± 0.22[a] | 9516 ± 1554[a] |
| Spring Valley | Ginger | 1 | 70175 ± 2565[a] | 3.24 ± 0.41[a] | 1367 ± 67[a] | 144331 ± 4122 | 5.87 ± 0.21[a] | 3203 ± 192[b] |
| Spring Valley | Ginger | 2 | 74395 ± 7388[a] | 2.72 ± 0.13[a] | 1521 ± 62[a] | 136009 ± 1325 | 4.31 ± 0.46[b] | 3741 ± 72[a] |
| Sundown Naturals | Ginseng Xtra | 1 | 9324 ± 423[a] | 1.09 ± 0.09[a] | 577 ± 16[a] | 4547 ± 399[a] | 0.73 ± 0.04[a] | 739 ± 31[a] |
| Sundown Naturals | Ginseng Xtra | 2 | 2202 ± 156[b] | 0.39 ± 0.05[b] | 0 ± 0[b] | 1524 ± 109[b] | 0.46 ± 0.04[b] | 138 ± 138[b] |
| Spring Valley | Korean Panax Ginseng | 1 | 146 ± 32[a] | 0.32 ± 0.04[a] | 1550 ± 255[a] | 9540 ± 261[a] | 0.40 ± 0.02[a] | 52 ± 14[a] |
| Spring Valley | Korean Panax Ginseng | 2 | 0 ± 0[b] | 0.05 ± 0.03[b] | 0 ± 0[b] | 0 ± 0[b] | 0.23 ± 0.03[b] | 0 ± 0[b] |
| Nature's Way | Red Raspberry Leaf | 1 | 614243 ± 29370[a] | 13.76 ± 0.30[a] | 8506 ± 112[a] | 73693 ± 1121[b] | 4.46 ± 0.33[a] | 3752 ± 53[b] |
| Nature's Way | Red Raspberry Leaf | 2 | 362000 ± 22822[b] | 11.00 ± 0.18[b] | 8038 ± 216[a] | 98551 ± 1702[a] | 4.14 ± 0.15[a] | 4410 ± 83[a] |
| Nature's Way | Reishi | 1 | 16681 ± 3489[a] | 2.87 ± 0.24[a] | 1784 ± 370? | 10549 ± 137[a] | 0.59 ± 0.04[a] | 3088 ± 830[a] |
| Nature's Way | Reishi | 2 | 14469 ± 415[a] | 1.72 ± 0.09[b] | 1154 ± 12 | 521 ± 82[b] | 0.53 ± 0.06[a] | 447 ± 23[b] |
| Nature's Way | Rhodiola | 1 | 859558 ± 40568[a] | 14.32 ± 0.67[a] | 16018 ± 439[a] | 451224 ± 109504[b] | 14.73 ± 0.67[b] | 15684 ± 1917[a] |
| Nature's Way | Rhodiola | 2 | 847358 ± 41650[a] | 16.86 ± 0.59[a] | 15114 ± 424[a] | 1063602 ± 134270[a] | 19.41 ± 1.27[a] | 21798 ± 1602[b] |
| NOW | Rhodiola | 1 | 739303 ± 57664[a] | 0.22 ± 0.04[b] | 5292 ± 512[b] | 45552 ± 4538[b] | 11.72 ± 1.43[b] | 8223 ± 386[b] |
| NOW | Rhodiola | 2 | 465838 ± 41714[b] | 14.97 ± 1.70[a] | 10235 ± 827[a] | 399962 ±23023[a] | 15.60 ± 1.01[a] | 13659 ± 1129[a] |
| Nature's Way | Silent Night | 1 | 87025 ± 2987[a] | 1.47 ± 0.03[b] | 6372 ± 256[a] | 10799 ± 1694 | 4.91 ± 0.11[a] | 2354 ± 70[a] |
| Nature's Way | Silent Night | 2 | 53731 ± 3951[b] | 5.70 ± 0.42[a] | 4277 ± 407[b] | 13289 ± 443 | 1.86 ± 0.10[b] | 675 ± 114[b] |
| Nature's Way | St. John's Wort | 1 | 166933 ± 3218[b] | 8.22 ± 0.37[a] | 9002 ± 360[a] | 66065 ± 1080[a] | 4.04 ± 0.11[a] | 4368 ± 161[a] |
| Nature's Way | St. John's Wort | 2 | 247036 ± 17717[a] | 7.79 ± 0.89[a] | 8509 ± 364[a] | 40182 ± 3011[b] | 4.24 ± 0.30[a] | 4188 ± 113[a] |
| Spring Valley | St. John's Wort | 1 | 90260 ± 8356[b] | 17.62 ± 1.25[a] | 16212 ± 1355[a] | 29367 ± 3146[b] | 17.58 ± 0.57[a] | 6862 ± 318[b] |
| Spring Valley | St. John's Wort | 2 | 271628 ± 10226[a] | 16.38 ± 0.56[a] | 15860 ± 559[a] | 158334 ± 9309[a] | 15.70 ± 0.59[b] | 13195 ± 335[a] |
| Sundown Naturals | St. John's Wort | 1 | 104370 ± 8184[b] | 15.56 ± 1.01[b] | 13471 ± 857[b] | 103772 ± 4655[b] | 12.37 ± 0.61[b] | 1115 ± 437[b] |
| Sundown Naturals | St. John's Wort | 2 | 287932 ± 18868[a] | 19.29 ± 0.56[a] | 20160 ± 1057[a] | 206181 ± 8777[a] | 15.18 ± 0.45[a] | 10539 ± 320[a] |
| Sundown Naturals | Stress formula | 1 | 39304 ± 1662[b] | 7.63 ± 0.34[a] | 10223 ± 335[a] | 24750 ± 220[a] | 0.76 ± 0.05[b] | 472 ± 30[b] |
| Sundown Naturals | Stress formula | 2 | 160332 ± 10585[a] | 5.43 ± 0.31[b] | 11174 ± 634[a] | 11586 ± 363[b] | 1.70 ± 0.45[a] | 1880 ± 52[a] |
| Nature's Way | Turmeric | 1 | 1555 ± 76[b] | 0.69 ± 0.04[a] | 0 ± 0[a] | 205518 ± 18668[a] | 14.51 ± 1.33[a] | 389853 ± 36119[a] |

*(Continued)*

**Table 3.** (Continued)

| Supplier | Supplement | Bottle | Water Extraction | | | Methanol Extraction | | |
|---|---|---|---|---|---|---|---|---|
| | | | A (TE) | P (GAE) | F (CE) | A (TE) | P (GAE) | F (CE) |
| Nature's Way | Turmeric | 2 | 1969 ± 119[a] | 0.67 ± 0.07[a] | 0 ± 0[a] | 161170 ± 5090[b] | 13.55 ± 0.59[a] | 289884 ± 8117[a] |
| Spring Valley | Turmeric | 1 | 4816 ± 1234[a] | 0.98 ± 0.10[a] | 5384 ± 1275[a] | 51320 ± 3681[b] | 1.69 ± 0.14[b] | 9548 ± 1036[b] |
| Spring Valley | Turmeric | 2 | 7081 ± 466[a] | 1.08 ± 0.07[a] | 824 ± 26[b] | 200538 ± 9945[a] | 15.44 ±1.25[a] | 89862 ± 7177[a] |
| Sundown Naturals | Turmeric | 1 | 16762 ± 520[a] | 0.86 ± 0.08[b] | 1236 ± 73[b] | 337434 ± 15251[a] | 18.26 ± 2.16[a] | 168328 ± 13461[a] |
| Sundown Naturals | Turmeric | 2 | 17944 ± 946[a] | 1.81 ± 0.23[a] | 2336 ± 333[a] | 195569 ± 7574[b] | 17.55 ± 0.73[a] | 184986 ± 7883[a] |
| Nature's Way | Valerian Root | 1 | 35271 ± 384[a] | 3.06 ± 0.07[a] | 3319 ± 154[a] | 13286 ± 770[a] | 1.41 ± 0.09[a] | 1929 ± 79[a] |
| Nature's Way | Valerian Root | 2 | 24052 ± 1366[b] | 2.48 ± 0.08[b] | 2893 ± 95[b] | 10081 ± 1148[b] | 1.53 ± 0.08[a] | 1687 ± 121[a] |
| Spring Valley | Valerian Root | 1 | 8208 ± 6651[a] | 0.52 ± 0.03[b] | 4091 ± 505[a] | 7993 ± 132[a] | 0.33 ± 0.01[b] | 105 ± 18[b] |
| Spring Valley | Valerian Root | 2 | 12573 ± 625[a] | 1.94 ± 0.25[a] | 1176 ± 65[b] | 4808 ± 180[b] | 1.25 ± 0.30[a] | 606 ± 24[a] |
| Sundown Naturals | Valerian Root | 1 | 53318 ± 1286[a] | 3.46 ± 0.11[a] | 3625 ± 147[a] | 17338 ± 997 | 1.59 ± 0.12[b] | 1796 ± 139[a] |
| Sundown Naturals | Valerian Root | 2 | 33033 ± 1876[b] | 4.30 ± 0.43[a] | 3224 ± 104[b] | 14606 ± 1054 | 2.04 ± 0.11[a] | 1913 ± 32[a] |
| Nature's Way | Yarrow | 1 | 114887 ± 12562[b] | 7.45 ± 0.91[a] | 11819 ± 1050[a] | 35094 ± 1356[b] | 2.69 ± 0.20[a] | 1840 ± 39[b] |
| Nature's Way | Yarrow | 2 | 167892 ± 12775[a] | 7.89 ± 0.98[a] | 10433 ± 809[a] | 39870 ± 641[a] | 2.74 ± 0.19[a] | 4360 ± 108[a] |

* The assays include the Ferrric Reducing Antioxidant Power (FRAP) assay for antioxidant capacity expressed as Trolox equivalents per gram, the Folin-Ciocalteu method for phenolics expressed as gallic acid equivalents per gram (GAE), and the AlCl3 precipitation assay for flavonoids expressed as +-catechin equivalents per gram. The values are expressed as the mean ± SE (n = 10) for each extraction from each bottle. Means within the same supplier for a supplement with different lowercase letters are significantly different based on LSD Student's T-Test (α of 0.05). For specific p-values for each supplement see S2 Table.

were completed (Table 5). A total decrease of greater than 25% in antioxidant capacity, phenolic content and flavonoid content was considered a significant change according to a 1-tailed t-test (S4 Table). Phenolic content was the most stable as the phenolic assay indicated a decreased in only 50% of the methanolic extracts, and only about 10% of the water extracts. Flavonoid concentration decreased in only 28% of the methanolic extracts, while it decreased in 100% of all of the water extracts. Antioxidant capacity demonstrated the least stability with a decrease in the majority of both the water and methanol extracts. The most stable supplement appeared to be supplements containing Echinacea.

## Metals and gross physical contamination

Zinc was the most abundant metal (Table 6), which was found in 88% of the bottles. Nickel was the next most common, found in about 40% of the samples. By contrast, lead (Pb) was found in none of the samples. Chromium was present in 5 bottles, but always at very low concentrations ($<0.05 \ \mu g \bullet g^{-1}$). Copper was found in the highest overall concentration of all the metals, but only in St. John's Wort from Spring Valley. The vast majority of bottles were uniform in texture and color, as observed under the dissecting microscope. There were a few notable exceptions. In one of the biotin bottles a thin black line of unknown origin was observed under the dissecting microscope, which was particularly notable because it is a bright white powder. In all the Turmeric bottles there was evidence of leaf particles in a few of the capsules; although, the capsules were listed as all being derived from the rhizome. Of the six bottles of Turmeric that were examined, one of the bottles was a dull yellow color rather than a bright orange and had very low values for all the assays (Fig 1), being significantly different from the other bottle from the same supplier. One other notable item is that batch number and the expiration date are completely absent from one of the bottles (Table 1) of Sundown Naturals Ginseng Xtra. The bottle was purchased at Walmart in Maryland at the same time as the purchase of the replicate bottle.

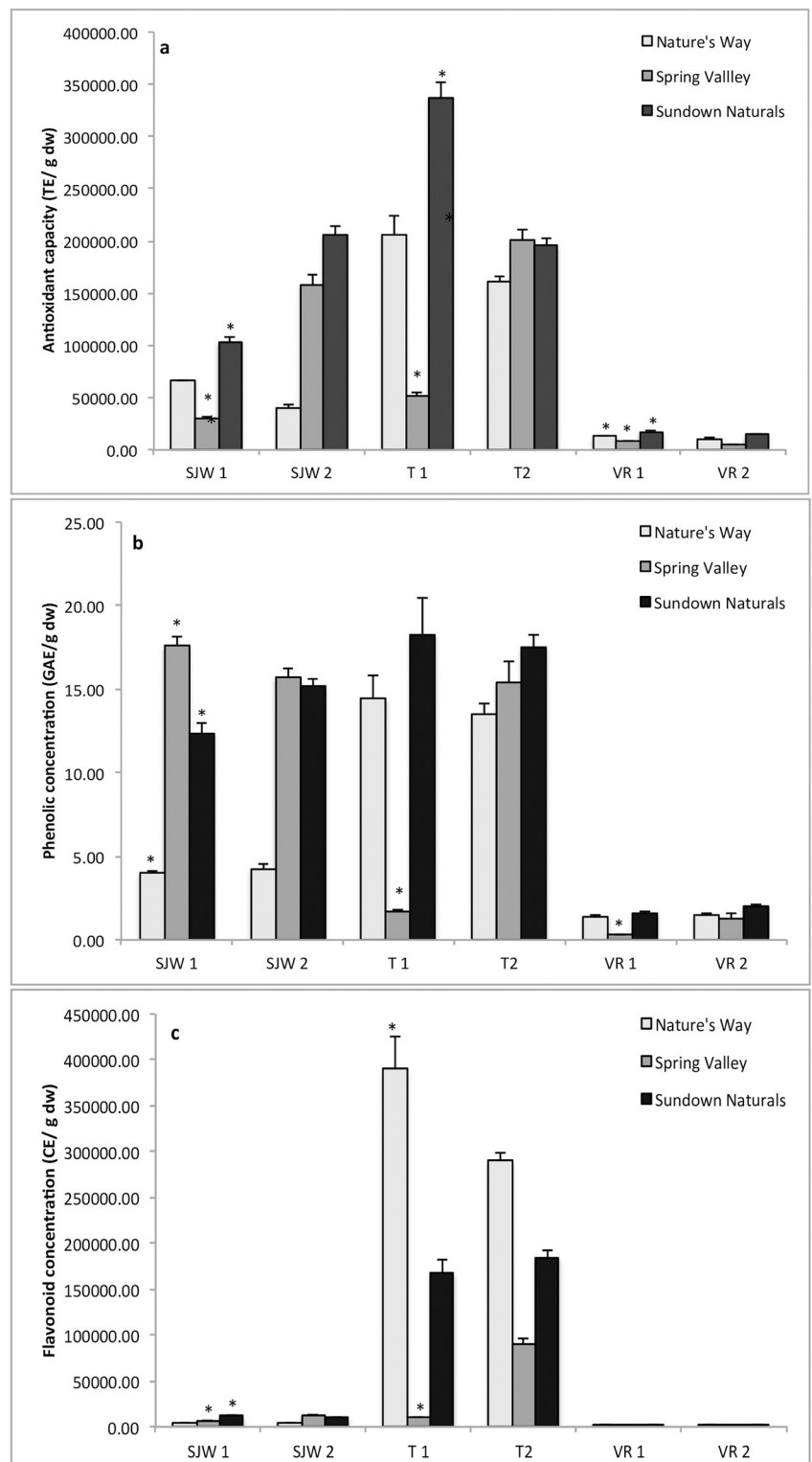

**Fig 1. The mean ± SE FRAP antioxidant capacity (a), phenolic concentration (b), and flavonoid concentration (c) of the methanolic fraction of two bottles each of St. John's Wort (SJW), Turmeric (T), and Valerian Root (VR) from three different suppliers.** The mean ± se from each bottle was calculated from 10 pills per bottle. An * indicates that bottle 1 is significantly different from bottle 2 according to ANOVA (p < 0.05).

**Table 4. Coefficient of variation (CV) for the FRAP antioxidant capacity (A), phenolic content (P) and flavonoid content (F) for 58 bottles of over the counter herbal supplements extracted with hot water and methanol.**

| Supplier | Supplement | Bottle | Water Extraction | | | Methanol Extraction | | |
|---|---|---|---|---|---|---|---|---|
| | | | A (TE)* | P (GAE) | F (CE) | A (TE) | P (GAE) | F (CE) |
| Nature's Way | Aloe | 1 | 34.27 | 31.93 | 55.40 | 9.87 | 22.60 | 18.89 |
| Nature's Way | Aloe | 2 | 6.29 | 4.23 | 2.67 | 11.03 | 21.01 | 13.85 |
| Nature's Way | Astragalus | 1 | 48.62 | 23.81 | 122.55 | 6.88 | 20.67 | 44.39 |
| Nature's Way | Astragalus | 2 | 15.00 | 13.39 | 11.93 | 66.72 | 7.85 | 11.25 |
| NOW | Astragalus | 1 | 9.37 | 38.60 | 7.23 | 12.69 | 15.35 | 9.24 |
| NOW | Astragalus | 2 | 9.34 | 27.84 | 9.94 | 18.00 | 14.53 | 18.24 |
| NOW | Biotin | 1 | 89.13 | 38.71 | 120.00 | 4.13 | 30.10 | 61.70 |
| NOW | Biotin | 2 | 22.23 | 18.81 | 0 | 161.93 | 25.24 | 0 |
| Nature's Way | Cranberry | 1 | 44.42 | 22.15 | 68.28 | 20.64 | 39.01 | 34.59 |
| Nature's Way | Cranberry | 2 | 9.79 | 8.39 | 10.65 | 14.22 | 26.90 | 43.47 |
| Spring Valley | Echinacea | 1 | 18.53 | 33.17 | 14.77 | 6.49 | 15.99 | 21.20 |
| Spring Valley | Echinacea | 2 | 59.95 | 14.16 | 8.21 | 7.38 | 27.26 | 8.90 |
| Sundown Naturals | Echinacea | 1 | 17.65 | 24.75 | 23.89 | 19.41 | 30.35 | 61.54 |
| Sundown Naturals | Echinacea | 2 | 7.89 | 29.93 | 25.39 | 18.06 | 187.57 | 57.86 |
| Nature's Way | Echinacea Goldenseal | 1 | 22.63 | 22.36 | 16.03 | 9.03 | 12.89 | 5.59 |
| Nature's Way | Echinacea Goldenseal | 2 | 9.07 | 9.59 | 9.27 | 2.53 | 47.92 | 23.05 |
| Spring Valley | Echinacea Goldenseal | 1 | 48.99 | 33.33 | 23.54 | 20.71 | 17.53 | 109.15 |
| Spring Valley | Echinacea Goldenseal | 2 | 11.57 | 25.67 | 34.89 | 11.02 | 18.91 | 78.70 |
| NOW | Ginger | 1 | 8.89 | 13.08 | 9.40 | 20.11 | 7.44 | 17.21 |
| NOW | Ginger | 2 | 9.93 | 22.92 | 14.38 | 14.37 | 9.24 | 51.64 |
| Spring Valley | Ginger | 1 | 11.56 | 40.34 | 15.49 | 9.03 | 11.37 | 18.94 |
| Spring Valley | Ginger | 2 | 31.40 | 15.15 | 12.96 | 3.08 | 33.48 | 6.07 |
| Sundown Naturals | Ginseng Xtra | 1 | 14.36 | 26.78 | 9.05 | 27.77 | 19.56 | 13.50 |
| Sundown Naturals | Ginseng Xtra | 2 | 21.23 | 39.37 | 0.00 | 22.53 | 24.74 | 300.00 |
| Spring Valley | Korean Panax Ginseng | 1 | 68.47 | 35.99 | 52.03 | 8.64 | 16.34 | 84.05 |
| Spring Valley | Korean Panax Ginseng | 2 | 0 | 169.19 | 0 | 0 | 40.26 | 0 |
| Nature's Way | Red Raspberry Leaf | 1 | 15.12 | 6.80 | 4.18 | 4.81 | 23.30 | 4.43 |
| Nature's Way | Red Raspberry Leaf | 2 | 19.94 | 5.26 | 8.49 | 5.46 | 11.82 | 5.95 |
| Nature's Way | Reishi | 1 | 66.15 | 26.32 | 65.56 | 4.10 | 21.29 | 85.03 |
| Nature's Way | Reishi | 2 | 9.08 | 15.65 | 3.33 | 49.90 | 37.46 | 16.58 |
| Nature's Way | Rhodiola | 1 | 14.92 | 14.74 | 8.67 | 76.74 | 14.49 | 38.66 |
| Nature's Way | Rhodiola | 2 | 15.54 | 11.11 | 8.88 | 39.92 | 20.68 | 23.25 |
| NOW | Rhodiola | 1 | 24.67 | 61.57 | 30.61 | 31.51 | 38.45 | 14.84 |
| NOW | Rhodiola | 2 | 28.32 | 35.93 | 25.55 | 18.20 | 20.38 | 26.13 |
| Nature's Way | Silent Night | 1 | 10.86 | 6.76 | 12.69 | 49.61 | 6.99 | 9.37 |
| Nature's Way | Silent Night | 2 | 23.26 | 23.57 | 30.14 | 10.54 | 17.10 | 53.23 |
| Nature's Way | St. John's Wort | 1 | 6.10 | 14.19 | 12.64 | 5.17 | 9.00 | 11.68 |
| Nature's Way | St. John's Wort | 2 | 22.68 | 36.18 | 13.53 | 23.70 | 22.51 | 8.52 |
| Spring Valley | St. John's Wort | 1 | 29.28 | 22.40 | 26.44 | 33.89 | 10.29 | 14.67 |
| Spring Valley | St. John's Wort | 2 | 11.91 | 10.84 | 11.15 | 18.59 | 11.85 | 8.03 |
| Sundown Naturals | St. John's Wort | 1 | 24.80 | 20.51 | 20.13 | 14.19 | 15.51 | 11.69 |
| Sundown Naturals | St. John's Wort | 2 | 20.72 | 9.14 | 16.58 | 13.46 | 9.42 | 9.60 |
| Sundown Naturals | Stress formula | 1 | 13.37 | 14.29 | 10.36 | 2.81 | 21.63 | 19.89 |
| Sundown Naturals | Stress formula | 2 | 18.67 | 17.96 | 17.95 | 9.91 | 84.20 | 8.70 |
| Nature's Way | Turmeric | 2 | 15.44 | 18.38 | 0.00 | 28.72 | 28.90 | 29.30 |

*(Continued)*

**Table 4.** (Continued)

| Supplier | Supplement | Bottle | Water Extraction | | | Methanol Extraction | | |
|---|---|---|---|---|---|---|---|---|
| | | | A (TE)* | P (GAE) | F (CE) | A (TE) | P (GAE) | F (CE) |
| Nature's Way | Turmeric | 1 | 19.19 | 31.13 | 0.00 | 9.99 | 13.82 | 8.85 |
| Spring Valley | Turmeric | 1 | 81.06 | 31.66 | 71.04 | 22.68 | 25.95 | 34.31 |
| Spring Valley | Turmeric | 2 | 20.81 | 21.45 | 10.13 | 15.68 | 25.50 | 25.26 |
| Sundown Naturals | Turmeric | 1 | 9.81 | 30.67 | 18.66 | 14.29 | 37.41 | 25.29 |
| Sundown Naturals | Turmeric | 2 | 16.68 | 40.59 | 45.14 | 12.25 | 13.10 | 13.48 |
| Nature's Way | Valerian Root | 1 | 3.45 | 7.56 | 14.70 | 18.34 | 19.56 | 12.90 |
| Nature's Way | Valerian Root | 2 | 17.96 | 9.84 | 10.37 | 36.01 | 17.17 | 22.74 |
| Spring Valley | Valerian Root | 1 | 256.21 | 15.81 | 39.06 | 5.23 | 12.28 | 54.16 |
| Spring Valley | Valerian Root | 2 | 15.71 | 41.21 | 17.44 | 11.86 | 75.11 | 12.73 |
| Sundown Naturals | Valerian Root | 1 | 7.63 | 10.20 | 12.86 | 18.19 | 23.35 | 24.48 |
| Sundown Naturals | Valerian Root | 2 | 17.96 | 31.90 | 10.20 | 22.82 | 16.68 | 16.32 |
| Nature's Way | Yarrow | 1 | 34.58 | 38.81 | 28.12 | 12.22 | 23.18 | 21.63 |
| Nature's Way | Yarrow | 2 | 24.06 | 39.45 | 24.53 | 5.09 | 21.86 | 7.82 |
| Rank Runs Test (p-values) | | | 0.114 | 0.114 | 0.993 | 0.993 | 0.008 | 0.002 |

* The assays include the Ferrric Reducing Antioxidant Power (FRAP) assay for antioxidant capacity expressed as Trolox equivalents per gram, the Folin-Ciocalteu method for phenolics expressed as gallic acid equivalents per gram (GAE), and the AlCl3 precipitation assay for flavonoids expressed as +-catechin equivalents per gram. The CV was based on a sample size of 10 pills from each bottle.

**Table 5. Change in the mean FRAP antioxidant capacity, phenolic and flavonoid concentration of two bottles each of nine herbal supplements*.**

| Supplier | Supplement | Bottle | Water Extractions | | | Methanol Extractions | | |
|---|---|---|---|---|---|---|---|---|
| | | | A(TE) | P (GAE) | F (CE) | A (TE) | P (GAE) | F (CE) |
| Spring Valley^ | Echinacea | 1 | | - | | | | |
| Spring Valley^ | Echinacea | 2 | - | | - | | | - |
| Sundown Naturals | Echinacea | 1 | | | - | - | | |
| Sundown Naturals | Echinacea | 2 | - | | - | - | | |
| Nature's Way | Echinacea Goldenseal | 1 | - | | - | - | - | |
| Nature's Way | Echinacea Goldenseal | 2 | - | | - | - | | |
| Spring Valley^ | Echinacea Goldenseal | 1 | | | - | - | - | |
| Spring Valley^ | Echinacea Goldenseal | 2 | | | - | - | | |
| Spring Valley | Turmeric | 1 | | | - | - | | - |
| Spring Valley | Turmeric | 2 | - | | - | - | | - |
| Sundown Naturals | Turmeric | 1 | - | | - | - | - | - |
| Sundown Naturals | Turmeric | 2 | - | - | - | - | | - |
| Nature's Way | Valerian | 1 | - | | - | - | | |
| Nature's Way | Valerian | 2 | - | | - | - | | |
| Spring Valley^ | Valerian | 1 | | | - | - | | |
| Spring Valley^ | Valerian | 2 | - | | - | - | | |
| Sundown Naturals | Valerian | 1 | - | | - | - | | |
| Sundown Naturals | Valerian | 2 | - | | - | - | | |

* Assays were conducted on two bottles each 5 capsules per bottle two years after the initial supplement testing conducted January to February 2019. Data was analyzed with a one-tailed t-test with an α level of 0.05 and a test value of < -25% change.

^Supplements that had expired at the time of the retest.

**Table 6. The mean ± SE concentrations of Ni, Cr, Cu, Zn and Pb in (μg • g$^{-1}$) isolated from 58 bottles of over the counter herbal supplements.**

| Supplier | Supplement | Bottle | Ni | Cr | Cu | Zn | Pb |
|---|---|---|---|---|---|---|---|
| Nature's Way | Aloe | 1 | 0 ± 0* | 0 ± 0 | 0 ± 0 | 2.65 ± 0.08 | 0 |
| Nature's Way | Aloe | 2 | 0.06 ± 0.05 | 0 ± 0 | 0 ± 0 | 0 ± 0 | 0 |
| Nature's Way | Astragalus | 1 | 0 ± 0 | 0 ± 0 | 0 ± 0 | 4.10 ± 2.98 | 0 |
| Nature's Way | Astragalus | 2 | 0 ± 0 | 0 ± 0 | 0 ± 0 | 0.71 ± 0.05 | 0 |
| NOW | Astragalus | 1 | 0 ± 0 | 0 ± 0 | 0 ± 0 | 0.68 ± 0.18 | 0 |
| NOW | Astragalus | 2 | 0 ± 0 | 0 ± 0 | 0 ± 0 | 0.38 ± 0.10 | 0 |
| NOW | Biotin | 1 | 0.11 ± .08 | 0 ± 0 | 0 ± 0 | 0.16 ± 0.08 | 0 |
| NOW | Biotin | 2 | 0 ± 0 | 0 ± 0 | 0 ± 0 | 0.69 ± 0.04 | 0 |
| Nature's Way | Cranberry | 1 | 0.45 ± 0.21 | 0 ± 0 | 0 ± 0 | 0.60 ± 0.19 | 0 |
| Nature's Way | Cranberry | 2 | 0.05 ± 0.05 | 0.04 ± 0.02 | 0 ± 0 | 1.06 ± 0.05 | 0 |
| Spring Valley | Echinacea | 1 | 0 ± 0 | 0 ± 0 | 0 ± 0 | 0.10 ± 0.07 | 0 |
| Spring Valley | Echinacea | 2 | 0.61 ± 0.34 | 0.03 ± 0.02 | 0 ± 0 | 0.34 ± 0.03 | 0 |
| Sundown Naturals | Echinacea | 1 | 0 ± 0 | 0 ± 0 | 0 ± 0 | 0.77 ± 0.07 | 0 |
| Sundown Naturals | Echinacea | 2 | 0 ± 0 | 0 ± 0 | 0 ± 0 | 0.75 ± 0.02 | 0 |
| Nature's Way | Echinacea Goldenseal | 1 | 0 ± 0 | 0 ± 0 | 0 ± 0 | 1.56 ± 0.08 | 0 |
| Nature's Way | Echinacea Goldenseal | 2 | 0.68 ± 0.31 | 0 ± 0 | 0 ± 0 | 1.19 ± 0.18 | 0 |
| Spring Valley | Echinacea Goldenseal | 1 | 0.02 ± 0.02 | 0 ± 0 | 0 ± 0 | 0.63 ± 0.07 | 0 |
| Spring Valley | Echinacea Goldenseal | 2 | 0 ± 0 | 0 ± 0 | 0 ± 0 | 0.32 ± 0.09 | 0 |
| NOW | Ginger | 1 | 0 ± 0 | 0 ± 0 | 0 ± 0 | 0.08 ± 0.03 | 0 |
| NOW | Ginger | 2 | 0 ± 0 | 0 ± 0 | 0 ± 0 | 0.08 ± 0.05 | 0 |
| Spring Valley | Ginger | 1 | 0 ± 0 | 0 ± 0 | 0 ± 0 | 1.01 ± 0.04 | 0 |
| Spring Valley | Ginger | 2 | 0 ± 0 | 0 ± 0 | 0 ± 0 | 1.20 ± 0.10 | 0 |
| Sundown Naturals | Ginseng Xtra | 1 | 0 ± 0 | 0 ± 0 | 0 ± 0 | 0 ± 0 | 0 |
| Sundown Naturals | Ginseng Xtra | 2 | 0 ± 0 | 0 ± 0 | 0 ± 0 | 0.07 ± 0.04 | 0 |
| Spring Valley | Korean Panax Ginseng | 1 | 0 ± 0 | 0 ± 0 | 0 ± 0 | 0.33 ± 0.23 | 0 |
| Spring Valley | Korean Panax Ginseng | 2 | 0 ± 0 | 0 ± 0 | 0 ± 0 | 0.08 ± 0.08 | 0 |
| Nature's Way | Red Raspberry Leaf | 1 | 0 ± 0 | 0 ± 0 | 0 ± 0 | 1.90 ± 0.04 | 0 |
| Nature's Way | Red Raspberry Leaf | 2 | 0 ± 0 | 0.42 ± 0.09 | 0 ± 0 | 0 ± 0 | 0 |
| Nature's Way | Reishi | 1 | 0 ± 0 | 0 ± 0 | 0 ± 0 | 0.23 ± 0.11 | 0 |
| Nature's Way | Reishi | 2 | 0.31± 0.20 | 0.17 ± 0.17 | 0 ± 0 | 5.15 ± 3.56 | 0 |
| Nature's Way | Rhodiola | 1 | 0 ± 0 | 0 ± 0 | 0 ± 0 | 0 ± 0 | 0 |
| Nature's Way | Rhodiola | 2 | 0.20 ± 0.12 | 0 ± 0 | 0 ± 0 | 0 ± 0 | 0 |
| NOW | Rhodiola | 1 | 0.13 ± 0.10 | 0 ± 0 | 0 ± 0 | 0.93 ± 0.80 | 0 |
| NOW | Rhodiola | 2 | 0 ± 0 | 0.08 ± 0.08 | 0 ± 0 | 2.29 ± 0.04 | 0 |
| Nature's Way | Silent Night | 1 | 0.23 ± 0.23 | 0 ± 0 | 0 ± 0 | 1.79 ± 0.40 | 0 |
| Nature's Way | Silent Night | 2 | 0 ± 0 | 0 ± 0 | 0 ± 0 | 1.51 ± 0.27 | |
| Nature's Way | St. John's Wort | 1 | 0 ± 0 | 0 ± 0 | 0 ± 0 | 1.00 ± 0.06 | 0 |
| Nature's Way | St. John's Wort | 2 | 0.26 ± 0.19 | 0 ± 0 | 0 ± 0 | 0.91 ± 0.06 | 0 |
| Spring Valley | St. John's Wort | 1 | 0 ± 0 | 0 ± 0 | 19.97 ± 0.85 | 1.93 ± 0.17 | 0 |
| Spring Valley | St. John's Wort | 2 | 0.01 ± 0.01 | 0 ± 0 | 15.00 ± 0.53 | 0.79 ± 0.05 | 0 |
| Sundown Naturals | St. John's Wort | 1 | 0.08 ± 0.08 | 0 ± 0 | 0 ± 0 | 0.78 ± 0.14 | 0 |
| Sundown Naturals | St. John's Wort | 2 | 0.18 ± 0.18 | 0 ± 0 | 0 ± 0 | 0.72 ± 0.08 | 0 |
| Sundown Naturals | Stress formula | 1 | 0.02 ± 0.02 | 0 ± 0 | 0 ± 0 | 1.02 ± 0.04 | 0 |
| Sundown Naturals | Stress formula | 2 | 0 ± 0 | 0 ± 0 | 0 ± 0 | 0.67 ± 0.16 | 0 |
| Nature's Way | Turmeric | 2 | 0 ± 0 | 0 ± 0 | 0 ± 0 | 0 ± 0 | 0 |
| Nature's Way | Turmeric | 1 | 0 ± 0 | 0 ± 0 | 0 ± 0 | 0 ± 0 | 0 |
| Spring Valley | Turmeric | 1 | 0.09 ± 0.06 | 0 ± 0 | 0 ± 0 | 1.69 ± 0.31 | 0 |

*(Continued)*

**Table 6.** (Continued)

| Supplier | Supplement | Bottle | Ni | Cr | Cu | Zn | Pb |
|---|---|---|---|---|---|---|---|
| Spring Valley | Turmeric | 2 | 0 ± 0 | 0 ± 0 | 0 ± 0 | 1.33 ± 0.23 | 0 |
| Sundown Naturals | Turmeric | 1 | 0 ± 0 | 0 ± 0 | 0 ± 0 | 0.08 ± 0.04 | 0 |
| Sundown Naturals | Turmeric | 2 | 0.01 ± 0.01 | 0 ± 0 | 0 ± 0 | 0.13 ± 0.05 | 0 |
| Nature's Way | Valerian Root | 1 | 0 ± 0 | 0 ± 0 | 0 ± 0 | 1.68 ± 0.11 | 0 |
| Nature's Way | Valerian Root | 2 | 0.16 ± 0.16 | 0 ± 0 | 0 ± 0 | 1.41 ± 0.08 | 0 |
| Spring Valley | Valerian Root | 1 | 0.12 ± 0.12 | 0 ± 0 | 0 ± 0 | 1.02 ± 0.12 | 0 |
| Spring Valley | Valerian Root | 2 | 0 ± 0 | 0 ± 0 | 0 ± 0 | 0.31 ± 0.11 | 0 |
| Sundown Naturals | Valerian Root | 1 | 0.01 ± 0.01 | 0 ± 0 | 0 ± 0 | 1.89 ± 0.05 | 0 |
| Sundown Naturals | Valerian Root | 2 | 0.08 ± 0.08 | 0 ± 0 | 0 ± 0 | 1.81 ± 0.09 | 0 |
| Nature's Way | Yarrow | 1 | 0.04 ± 0.04 | 0 ± 0 | 0 ± 0 | 3.44 ± 0.95 | 0 |
| Nature's Way | Yarrow | 2 | 0.09 ± 0.09 | 0 ± 0 | 0 ± 0 | 0.96 ± 0.19 | 0 |
| *Bottles where element present* | | | 24/58 | 5/58 | 2/58 | 51/58 | 0/58 |

*The values are expressed as the mean ± SE (n = 5) for each extraction from each bottle.

## Fungal isolation

All control plates were clear of fungal growth. Four of the tested supplements were completely clear of fungal growth but were not isolated to any one supplier (Table 7). Of the bottles tested, 37 of the 58 had fungal contamination, with 21 of the supplements containing multiple microbial isolates, ranging from two to six species. The types of fungi that were isolated from the samples varied, but the most common fungi were in the genus *Aspergillus*. Other commonly isolated fungi include species from the genus *Candida*, *Microsporum*, and *Nocardia*. In addition, there was some bacterial growth on the SDA plates, such as *Bacillus* and *Streptomyces*.

# Discussion

## Assays and supplement stability

Recognizing that there are multiple methods of measuring antioxidant capacity, phenolics and flavonoids it was decided that one type of assay for each should be conducted to more quickly determine if there is justification for further investigation into the purity and consistency of the supplements. An important next step is testing the supplements for the presence of and concentration of the active ingredient if present. The data included in this analysis, in conjunction with previous studies on supplement contamination, strengthen the case that the FDA should regulate over-the-counter herbal supplements the same way that they regulate food and drugs and that further testing involving different assays for antioxidant capacity and specific phenolics and flavonoids is justified for these supplements. The inconsistency between different bottles of the same supplement from the same supplier (as demonstrated by the large standard errors) demonstrates that there seems to be little consistency in the production of herbal supplements among all the suppliers that were tested. Bottles from the same batch theoretically should have low differences, as they should have been produced side by side with the same manufacturing process [2]. Of the 18 bottles that were retested after two years, 6 were past their expiration date, while 12 were well within their use by date, so it was anticipated that there would be less change for these supplements, but this was not observed. There was no consistent pattern displayed by bottles within their use by date versus those that had expired. When there was a significant difference between the supplements that had expired and those that had not in all but one case the expired supplement had more stability than the non-

**Table 7. Microbial species identified from four samples from each of two bottles from 29 herbal supplements.** Isolates are separated by supplier and supplement type.

| Nature's Way | Bottles | Identified Contaminant |
|---|---|---|
| Aloe | 2/2 | *Aspergillus sp., Sporobolomyces salmonicolor, Trichophyton sp.* |
| Astragalus | 1/2 | *Bacillus subtilis, Candida tropicalis* |
| Cranberry | 1/2 | *Aspergillus fumig\atus, Microsporum sp., Trichophyton terrestre* |
| Echinacea | 2/2 | *Candida albicans, Sporobolomyces, Microsporum sp.* |
| Echinacea Goldenseal | 2/2 | *Bacillus subtilis, Rhodotorula* |
| Red Raspberry Leaf | 2/2 | *Aspergillus fumigatus, Microsporum sp., Nocardia brasiliensis* |
| Reishi | 1/2 | *Bacillus subtilis* |
| Rhodiola | 2/2 | *Sporobolomyces salmonicolor* |
| Silent Night | 1/2 | *Candida sp., Candida tropicalis, Nocardia brasiliensis, Streptomyces* |
| St. John's Wort | 0/2 | *none* |
| Turmeric | 1/2 | *Chlorociboria aeruginascens* |
| Valerian Root | 2/2 | *Aspergillus sp., Bacillus subtilis, Crytococcus neoformans, Candida krusei, Nocardia brasiliensis* |
| Yarrow | 0/2 | *none* |
| **NOW** | | |
| Astragalus | 2/2 | *Aspergillus fumigatus, Aspergillus sp., Bacillus subtilis, Sporobolomyces sp., Sporotrichum sp., Trichophyton rubrum* |
| Biotin | 1/2 | *Lycogala epidendrum, Rhodotorula sp.* |
| Ginger | 2/2 | *Acremonium sp., Bacillus subtilis, Chromelosporium fulva, Cryptococcus sp., Nocardia brasiliensis, Sporobolomyces salmonicolor* |
| Rhodiola | 0/2 | *none* |
| **Spring Valley** | | |
| Echinacea | 1/2 | *Rhodotorula sp., Streptomyces sp.* |
| Echinacea Goldenseal | 2/2 | *Aspergillus sp., Beauveria sp., Trichophyton sp.* |
| Ginger Root | 1/2 | *Bacillus subtilis, Microsporum sp., Nocardia brasiliensis, Sporobolomyces salmonicolor* |
| Korean Panax Ginseng | 1/2 | *Cryptococcus neoformans* |
| St. John's Wort | 1/2 | *Aspergillus niger, Aspergillus sp., Cryptococcus sp.* |
| Turmeric | 1/2 | *Aspergillus sp., Candida tropicalis, Penicillium sp., Nocardia brasiliensis* |
| Valerian Root | 2/2 | *Aspergillus fumigatus, Aspergillus niger, Candida sp., Cryptococcus sp., Pseudomonas sp.* |
| **Sundown Naturals** | | |
| Echinacea | 1/2 | *Bacillus subtilis, Nocardia brasiliensis* |
| Ginseng Xtra | 1/2 | *Aspergillus flavus, Bacillus subtilis, Nocardia brasiliensis, Rhizopus sp.* |
| St. John's Wort | 1/2 | *Aspergillus fumigatus* |
| Stress Formula | 2/2 | *Bacillus subtilis, Cryptococcus sp., Nocardia brasiliensis, Trichophyton sp.* |
| Turmeric | 0/2 | *none* |
| Valerian Root | 1/2 | *Aspergillus fumigatus, Bacillus subtilis, Candida sp., Nocardia brasiliensis, Trichophyton terrestre* |

expired supplement (data not shown). Valerian root that had expired had a more significant decrease in flavonoid content in the water soluble fraction; however, all bottles saw a decrease of more than 25% (data not shown). The least stable values between the two testing periods were antioxidant activity in both the methanolic and water extracts and the flavonoid content in the water extract (Table 4). Phenolic content in both extracts and flavonoid content of the methanol extract were the most stable after two years (Table 4). The stability of antioxidant

capacity can be susceptible to temperature, light and humidity, depending on the types of anti-oxidants in the supplements [42]. If this was the case in these data, which mimicked storage conditions in a home, consumers should be made aware that stability and efficacy of the products could be impacted by storage conditions.

## Metal analysis

In addition to issues with consistency within and between bottles, was the data associated with the presence of metals and microbial isolates in the supplements. The recommended daily allowance (RDA) for adults for copper is 700 µg per day with an upper limit of 10 mg per day [43]. The RDA for adults for zinc is 15 mg per day with an upper limit of 40 mg per day [43]. The RDA for adults for chromium is 24 µg per day with no established upper limit due to lack of research [39]. The upper limit for nickel for adults is 1 mg per day [43]. These RDA values are usually met by taking a single dose of a daily multivitamin and most often will be met by eating a well-balanced diet. Surpassing the upper limit of minerals is not typically through diet, but by ingesting contaminated water supplies or from their presence within supplements [43, 44]. The main risk factor leading to metal overdose is that people who take supplements often take multiple supplements at the same time [45–47]. Overconsumption of these supplements when taken together, particularly over a prolonged period, can lead to excess zinc, copper, chromium and nickel build up, which could lead to mineral toxicity.

The metal screens indicated a low concentration of metals in general; apart from copper from St. John's Wort produced by Spring Valley (Table 6), with values that ranged from 13–22 µg • g$^{-1}$. High levels of copper intake can lead to copper toxicity, which has been implicated in a few psychiatric disorders such as Alzheimer's and Parkinson's [48]. In addition, high copper intake in conjunction with a high fat diet has been linked to reductions in cognition in the general population [46]. This is particularly problematic with the high copper levels found in St. John's Wort from Spring Valley. Although the amount of copper within the supplement is most likely not problematic for the majority of the population, it could unintentionally lead to dangerous copper levels in individuals who already have higher than normal copper uptake or susceptible populations. The St. John's Wort bottles from Spring Valley, furthermore, were from different batch numbers, which indicates that this may be a chronic issue with the supplement supply or with the processing of the supplement. High amounts of dietary zinc, which could be exacerbated by taking multiple supplements, alters intestinal bacteria and can increase susceptibility to *Clostridium difficile* infection [49]. Caution is indicated with consumption of supplements given the above findings because other minerals that were not tested may also be present in supplements, which may have as of yet unknown and unintended negative health consequences.

## Fungal isolation

About 80% of the microbial screens were contaminated with fungus, with a few bacteria showing up in the fungal screens. Morphological identification of the species indicates that there were potential species of concern, which indicate targets for further research to determine if the species tentatively identified are indeed problematic. Of the microbial isolates that were classified only one, *Aspergillus flavus*, is known to produce mycotoxins [50]. Isolates from three genera, *Candida*, *Trichophyton* and *Microsporum* are known to cause infections and can be particularly problematic for immunocompromised individuals [51–53]. Most of the isolates that were identified, such as *Nocardia*, *Rhodotorula*, and *Sporobolomyces*, are generally safe, but may act as opportunistic pathogens in immunocompromised individuals [54–56]. Other isolates, such as *Chlorociboria aeruginascens*, *Lycogala epidendrum*, *Chromelosporium fulva*,

*Beauveri* and *B. subtilis* generally have no impact on human health as noted by a search of the literature.

## Conclusion

Due to the high demand for herbal supplements and other natural products, suppliers should follow strict manufacturing practices that ensure consistency, purity, and safety of their products. The data presented here strengthens the case that lack of regulation leads to products that are not standardized and have the potential to either be ineffective or to cause harm. Currently in the United States manufacturers are to ensure safety of their products, but the FDA can only legally act to remove supplements when there is proof of harm, false statements or ineffective labelling [57]. Until the regulation of herbal supplements falls under the jurisdiction of the FDA, the general public should be informed that if they are consuming herbal supplements they may not be getting what they are paying for.

## Supporting information

**S1 Table. P-values of the analysis of herbal supplements from two or more suppliers.** P-values are based on ANOVA with an α level of 0.05.
(PDF)

**S2 Table. P-values of the analysis of bottles within a supplier for supplement.** P-values are based on ANOVA with an α level of 0.05.
(PDF)

**S3 Table. Range of coefficient of variations (CV) from 58 bottles of over the counter herbal supplements.** Antioxidant capacity, phenolic concentrations, and flavonoid concentrations were measured from both hot water and methanolic extractions of 10 pills from each of two bottles per supplement per supplier. Results of the Rank Runs test are also included with p-value of $< 0.05$. indicating non-randomness of distribution among the results.
(PDF)

**S4 Table. P-value from a 1 tailed t-test.** T-tests indicated a change in antioxidant capacity, phenolic and flavonoid concentration of -25% from the original test.
(PDF)

## Author Contributions

**Conceptualization:** Maren E. Veatch-Blohm, Iris Chicas, Kathryn Margolis, Rachael Vanderminden.

**Data curation:** Maren E. Veatch-Blohm, Kathryn Margolis.

**Formal analysis:** Maren E. Veatch-Blohm, Iris Chicas, Marisa Gochie, Khusmanie Lila.

**Investigation:** Maren E. Veatch-Blohm, Iris Chicas, Kathryn Margolis, Rachael Vanderminden, Marisa Gochie, Khusmanie Lila.

**Methodology:** Maren E. Veatch-Blohm.

**Project administration:** Maren E. Veatch-Blohm.

**Resources:** Maren E. Veatch-Blohm.

**Supervision:** Maren E. Veatch-Blohm.

**Writing – original draft:** Maren E. Veatch-Blohm, Iris Chicas, Kathryn Margolis, Rachael Vanderminden.

**Writing – review & editing:** Maren E. Veatch-Blohm, Iris Chicas, Kathryn Margolis, Marisa Gochie, Khusmanie Lila.

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
