## [Decision Letter · Decision Letter 0]

14 Sep 2020

PONE-D-20-06630

Screening for consistency and contamination within and between bottles of 29 herbal supplements.

PLOS ONE

Dear Dr. Maren,

Thank you for submitting your manuscript to PLOS ONE. After careful consideration, we feel that it has merit but does not fully meet PLOS ONE’s publication criteria as it currently stands. Therefore, we invite you to submit a revised version of the manuscript that addresses the points raised during the review process.

ACADEMIC EDITOR:

The manuscript requires major revision to address concerns raised by reviewers.

Kindly give emphasis on the following during revisions:

1. Conduct thorough statistical analysis for data presented in the results section to make better interpretations of the findings.

2. Insert missing materials and methods section using standard protocols and give well described methodology of how the data was collected for all the parameters analyzed.

3. Include supplementary data that can add quality to the findings if you have more data pending that wasn't included in the manuscript, this increase quality of the publication.

4. Improve the discussion section to be in line with the results stated in a logical sequence.

5. Check and correct the overall quality of the manuscript based on the reviewers' comments including missing sections and editorial/grammatical errors.

We look forward to receiving your revised manuscript.

Kind regards,

Catherine Nkirote Kunyanga, Ph.D.

Academic Editor

PLOS ONE

Journal Requirements:

Reviewers' comments:

Reviewer's Responses to Questions

**Comments to the Author**

1. Is the manuscript technically sound, and do the data support the conclusions?

Reviewer #1: No

2. Has the statistical analysis been performed appropriately and rigorously? 

Reviewer #1: No

3. Have the authors made all data underlying the findings in their manuscript fully available?

Reviewer #1: Yes

4. Is the manuscript presented in an intelligible fashion and written in standard English?

Reviewer #1: No

5. Review Comments to the Author

Reviewer #1: It is a very good initiation to work on ensuring the consistency, purity, and safety of herbs relating to heavy metals and microorganisms, which have a role in adaptogens, stress and depression relief, and immune response. However, I have doubtful to consider this manuscript as a full research paper. If there is a room, PLOS can also handle as a short communication or letter.

In addition, I do have the following comment.

Line 21-23, Please state the quantitative value/data (mean±sd) for antioxidant activity, phenolic concentrations, and flavonoid concentrations.

Line 15. Please put in bracket (FDA) after the Food and Drug Administration (FDA), as this is the 1st appearance.

Line 33 Key Words, please make Keywords, please also check for the number of the keywords included, from PLOS authors guide.

Line 99. Please make subsections for Materials and Methods.

Example

Sample collection

Laboratory analysis

Sample extraction

Antioxidant

Stability test

Metals test

Physical contamination

Fungi test

Please also make the subsections for your Discussion; based on the content of your Discussion.

The authors repeatedly use personal pronouns (we) in a different part of the manuscript that needs to be changed/modified, Line 101, 102, 104 122, 141, 152, 158, 169, 180, 201, 204, 219, 224, 241, 245, 282, 285.

Line 107, the title of Table 1 is lengthy, please revise, and if need be, put all the other descriptions at the bottom of the table in the bracket or asterisk.

Line 113 please indicate the source/reference for extraction, which consists of the detailed protocol.

Line 118 state the source, at the end of the sentence.

Line 119. FRAP please write Ferric reducing antioxidant power and FRAP in the bracket.

Line 119. For antioxidant in vitro test, at least two assays needed; in addition, better also to look the Effective concentration (EC50). These make the antioxidant activity test/data to the standard and trustful.

Line 130, please state a source for the physical contamination test.

Line 139 and 140 please state a source.

Line 148,149, please state the alpha level for the degree of significance.

Line 170. Table 2 needs a general modification.

This table should be with the numerical value (mean±se for each supplement and parameters). The mean values should be also differentiated using superscript letters if significant differences found.

Line 170. Please state the title of Table 2, and put the others, as the footnote at the bottom of the Table.

Line 188 please rewrite the title of Table 3.

Table 3. Better to describe as mean±sd, please.

Table 3. needs to have superscript letters to show the degree of significance.

Table 3. An antioxidant is a broad term that does not described as a single parameter. It needs to be specific. In your case, FRAP. Please know that a single in vitro parameter is not to the standard to support the antioxidant activity of your samples. Please refer to the comment give in Line 119.

Line 209 Table 4, please modify the title of the Table, state mean±sd with significant superscript letters. The range value can be indicated in your discussion.

Line 237, Our data, in conjunction with previous studies on supplement contamination, …..

Please substitute Our data, ……

The findings of this study/data

Please apply these changes, wherever needed.

Line 281 please apply the above comment (Line 237).

Page 24 I am not clear for the question mark found in Figure 1.

Thank you

Abera Belay

6. PLOS authors have the option to publish the peer review history of their article (what does this mean?). If published, this will include your full peer review and any attached files.

Reviewer #1: No

---

## [Author Response · Author response to Decision Letter 0]

29 Sep 2020

Reviewer #1: No – Actually the manuscript is written in standard English; although, grammatical issues related to use of first person rather than passive voice have been addressed. However, some of the reviewer’s comments were not written using correct English grammar and syntax.

5. Review Comments to the Author

1. Reviewer #1: It is a very good initiation to work on ensuring the consistency, purity, and safety of herbs relating to heavy metals and microorganisms, which have a role in adaptogens, stress and depression relief, and immune response. However, I have doubtful to consider this manuscript as a full research paper. If there is a room, PLOS can also handle as a short communication or letter. – Hopefully the addition of the complete statistical analysis will allow the paper to be considered as a full research paper.

2. Line 21-23, Please state the quantitative value/data (mean±sd) for antioxidant activity, phenolic concentrations, and flavonoid concentrations. – Included one the range of CVs as an example, but it would be impractical to include the quantitative data for all of the supplements in a meaningful way within the abstract of the manuscript. In this case the CV is ideal as it gives an idea of variability that is standardized across values that are different by orders of magnitude.

3. Line 15. Please put in bracket (FDA) after the Food and Drug Administration (FDA), as this is the 1st appearance. - Done

4. Line 33 Key Words, please make Keywords, please also check for the number of the keywords included, from PLOS authors guide. – There are already keywords included in the paper. I could not find information about key words in the authors guide

5. Line 99. Please make subsections for Materials and Methods.

Example

Sample collection

Laboratory analysis

Sample extraction

Antioxidant

Stability test

Metals test

Physical contamination

Fungi test

Please also make the subsections for your Discussion; based on the content of your Discussion. – Done, the number of subsections included is not as numerous as suggested in the above example as this would have impeded the flow of the paper. However, there are now subsections for the Materials and Methods, Results and Discussion 

6. The authors repeatedly use personal pronouns (we) in a different part of the manuscript that needs to be changed/modified, Line 101, 102, 104 122, 141, 152, 158, 169, 180, 201, 204, 219, 224, 241, 245, 282, 285. – Done

7. Line 107, the title of Table 1 is lengthy, please revise, and if need be, put all the other descriptions at the bottom of the table in the bracket or asterisk. -Done

Line 113 please indicate the source/reference for extraction, which consists of the detailed protocol. – Done - Cai, Y., Luo, Q., Sun, M., Corke, H., 2004. Antioxidant activity and phenolic compounds of 112 traditional Chinese medicinal plants associated with anticancer. Life Sci. 74, 2157-2184. 

8. Line 118 state the source, at the end of the sentence. – In the original manuscript only line 118 refers to an extraction protocol, while line 113 is the title for table 1. So I’ve included the source for the extraction protocol, but there was no source for what supplements were tested. These were chosen by the lab group and the subsampling was done to ensure that each pill had all 4 extractions conducted. 

9. Line 119. FRAP please write Ferric reducing antioxidant power and FRAP in the bracket. - Done

10. Line 119. For antioxidant in vitro test, at least two assays needed; in addition, better also to look the Effective concentration (EC50). These make the antioxidant activity test/data to the standard and trustful. – The reason for choosing only one type of assay for each type is included at the beginning of the discussion. “Recognizing that there are multiple methods of measuring antioxidants, phenolics and flavonoids it was decided that one type of assay for each should be conducted to more quickly determine if there is justification for further investigation into the purity and consistency of the supplements.”

11. Line 130, please state a source for the physical contamination test. – There is no source for this. We just examined the samples under the dissecting microscope to see if anything looked unusual or out of place.

12. Line 139 and 140 please state a source. – The source was added. Singh P, Srivastava B, Kumar A, Dubey NK. Contamination of raw materials of some herbal drugs and recommendation of cinnamomum camphora oil as herbal fungitoxicant. Micro Ecol. 2008; 56: 555-560.

13. Line 148,149, please state the alpha level for the degree of significance. - Done

14. Line 170. Table 2 needs a general modification.

This table should be with the numerical value (mean±se for each supplement and parameters). The mean values should be also differentiated using superscript letters if significant differences found. – This data is now split between 2 tables. One based on the means and SE for each bottle of each supplement and supplier. The other is based on the overall means for each supplier of each supplement (now tables 2 and 3)

15. Line 170. Please state the title of Table 2, and put the others, as the footnote at the bottom of the Table. - Done

16. Line 188 please rewrite the title of Table 3. - Done

17. Table 3. Better to describe as mean±sd, please. – The table has been changed to show the CV for all values, but we believe that the CV is valuable in evaluating supplements with a wide range of values. Therefore we are including an expanded CV table in addition to the means and standard errors with accompanying superscripts in two new tables. See 15 above.

18. Table 3. needs to have superscript letters to show the degree of significance. - Done

19. Table 3. An antioxidant is a broad term that does not described as a single parameter. It needs to be specific. In your case, FRAP. Please know that a single in vitro parameter is not to the standard to support the antioxidant activity of your samples. Please refer to the comment give in Line 119. – See response to comment 11 above.

20. Line 209 Table 4, please modify the title of the Table, state mean±sd with significant superscript letters. The range value can be indicated in your discussion. – The title has been modified

21. Line 237, Our data, in conjunction with previous studies on supplement contamination, …..

Please substitute Our data, ……

The findings of this study/data

Please apply these changes, wherever needed. - Done

22. Line 281 please apply the above comment (Line 237). – Done

23. Page 24 I am not clear for the question mark found in Figure 1. – I have carefully looked over the manuscript figure file and found no question mark in Figure 1.

---

## [Decision Letter · Decision Letter 1]

24 Nov 2020

PONE-D-20-06630R1

Screening for consistency and contamination within and between bottles of 29 herbal supplements.

PLOS ONE

Dear Dr. Veatch-Blohm,

Thank you for submitting your manuscript to PLOS ONE. After careful consideration, we feel that it has merit but does not fully meet PLOS ONE’s publication criteria as it currently stands. Therefore, we invite you to submit a revised version of the manuscript that addresses the points raised during the review process.

In addition to responding the points raised by the reviewer, please address the following items:

We note that line 68 is not supported by citation to a reference, but this could be because it may be intended to set up the reference in line 76. Please confirm or revise as appropriate.  As currently phrased, it could reads like manufacturers intentionally contaminated supplements. Is this what the authors intended to say?  If not, perhaps revise like this: “Contamination may be due to growth, processing or storage conditions or may be intentional addition of unlisted ingredients.”Lns 83-86:  Please provide links to FDA sources describing the three 2015 recalls reported. In the Materials and Methods section, please specify exactly when the testing was done.Lns 184-85:  Please clarify whether any differences reported were statistically significant.Lns 237-38 & 297-306: Is the suggestion that ingesting copper levels detected in Spring Valley’s St. John’s Wort over time result in copper toxicity bad enough to cause Alzheimer’s, Parkinson’s or cognitive decline? The current phrasing seems to imply this is the case, even when taken a “correct dosage” (at least with respect to cognitive decline). This should be substantiated.Lns 244-48:  Please indicate where the bottle described in these lines was purchased to avoid any argument that the bottle tested was not a genuine bottle but a counterfeit.  

We look forward to receiving your revised manuscript.

Kind regards,

Jamie Males, Ph.D.

Senior Editor

PLOS ONE

on behalf of

Catherine Nkirote Kunyanga, Ph.D.

Academic Editor

PLOS ONE

Reviewers' comments:

Reviewer's Responses to Questions

**Comments to the Author**

1. If the authors have adequately addressed your comments raised in a previous round of review and you feel that this manuscript is now acceptable for publication, you may indicate that here to bypass the “Comments to the Author” section, enter your conflict of interest statement in the “Confidential to Editor” section, and submit your "Accept" recommendation.

Reviewer #1: (No Response)

2. Is the manuscript technically sound, and do the data support the conclusions?

Reviewer #1: No

3. Has the statistical analysis been performed appropriately and rigorously? 

Reviewer #1: No

4. Have the authors made all data underlying the findings in their manuscript fully available?

Reviewer #1: No

5. Is the manuscript presented in an intelligible fashion and written in standard English?

Reviewer #1: No

6. Review Comments to the Author

Reviewer #1: This is my 2nd review for the authors, please.

Hopefully, the authors will consider all the comments to maintain the quality of the manuscript for the readership of the PLOS community.

As a reviewer, I am still sticking to my previous recommendation. Additional data is necessary to consider this paper as a full research manuscript, and need to improve and thoroughly discuss the tables and figures.

2. Line 21-23, Please state the quantitative value/data (mean±sd) for antioxidant

activity, phenolic concentrations, and flavonoid concentrations.

– Included one the range of CVs as an example, but it would be impractical to include the quantitative data for all of the supplements in a meaningful way within the abstract of the manuscript. In this case the CV is ideal as it gives an idea of variability that is standardized across values that are different by orders of magnitude.

My request is to include the mean±sd of antioxidant activity, phenolic concentrations, and flavonoid concentrations in the abstract section. The authors do not address this.

The description of statistical variation needs to be based on mean±sd, and the superscript should be stated for each treatment, which is based on the treatment variation. (This is also seen in the new document, line number 163 page 10).

10.Line 119. For antioxidant in vitro test, at least two assays needed; in addition, better also to look the Effective concentration (EC50). These make the antioxidant activity test/data to the standard and trustful.

– The reason for choosing only one type of assay for each type is included at the beginning of the discussion. “Recognizing that there are multiple methods of measuring antioxidants, phenolics and flavonoids it was decided that one type of assay for each should be conducted to more quickly determine if there

is justification for further investigation into the purity and consistency of the

supplements.”

Phenol and flavonoids are phytochemicals, which are considered as antioxidant content. My request is for antioxidant activity, like FRAP. In line with this calculating EC50 increases the readability of the manuscript, and also makes the data trustful.

13. Line 148,149, please state the alpha level for the degree of significance.

- Done

The authors said done for this comment; however, the alpha level for the degree of significance is not in place.

14. Line 170. Table 2 needs a general modification. This table should be with the numerical value (mean±se for each supplement and parameters). The mean values should be also differentiated using superscript letters if significant differences are found.

– This data is now split between 2 tables. One based on the means and SE for each bottle of each supplement and supplier. The other is based on the overall means for each supplier of each supplement (now tables 2 and 3).

The superscript stated in Table 2 and 3 are not correct, and is not statistical accepted. Some of the values do not have also the superscripts.

17.Table 3. Better to describe as mean±sd, please.

– The table has been changed to show the CV for all values, but we believe that the CV is valuable in evaluating supplements with a wide range of values. Therefore we are including an expanded CV table in addition to the means and standard errors with accompanying superscripts in two new tables. See 15 above.

To my knowledge, CV does not indicate a statistical variation between treatments. For a reputable journal like PLOS ONE, a description of treatments using a statistical variation is important for the readers, and maintain the quality of the journal.

18. Table 3. needs to have superscript letters to show the degree of significance. –Done.

This is not properly done.

19. Table 3. An antioxidant is a broad term that does not described as a single

parameter. It needs to be specific. In your case, FRAP. Please know that a single in vitro parameter is not to the standard to support the antioxidant activity of your samples. Please refer to the comment give in Line 119. – See response to comment 11 above.

This is not addressed.

20.Line 209 Table 4, please modify the title of the Table, state mean±sd with significant superscript letters. The range value can be indicated in your discussion.

– The title has been modified.

The title is modified; however, it is necessary to address the other requests.

Thank you

7. PLOS authors have the option to publish the peer review history of their article (what does this mean?). If published, this will include your full peer review and any attached files.

Reviewer #1: No

---

## [Author Response · Author response to Decision Letter 1]

2 Jan 2021

Response to Reviewer comments

• We note that line 68 is not supported by citation to a reference, but this could be because it may be intended to set up the reference in line 76. Please confirm or revise as appropriate. As currently phrased, it could reads like manufacturers intentionally contaminated supplements. Is this what the authors intended to say? If not, perhaps revise like this: “Contamination may be due to growth, processing or storage conditions or may be intentional addition of unlisted ingredients.” – The wording has been changed as per your suggestion. The sentence was to provide a set up for the rest of the paragraph.

• Lns 83-86: Please provide links to FDA sources describing the three 2015 recalls reported. – The archived links are now included in the reference section under reference 28.

• In the Materials and Methods section, please specify exactly when the testing was done. – The following statement was added at the end of the first paragraph of the materials and methods. – “ All of the extractions and tests, with the exception of the supplement stability tests were conducted from October 2016 through April 2017. The supplement stability tests were conducted in January and February 2019.”

• Lns 184-85: Please clarify whether any differences reported were statistically significant. – The wording has been changed to show that differences reported among bottles indicate a significant difference.

• Lns 237-38 & 297-306: Is the suggestion that ingesting copper levels detected in Spring Valley’s St. John’s Wort over time result in copper toxicity bad enough to cause Alzheimer’s, Parkinson’s or cognitive decline? The current phrasing seems to imply this is the case, even when taken a “correct dosage” (at least with respect to cognitive decline). This should be substantiated. – The wording here has been changed to the following, which hopefully removes a causalconnection – “Although the amount of copper within the supplement is most likely not problematic for the majority of the population, it could unintentionally lead to dangerous copper levels in individuals who already have higher than normal copper uptake or susceptible populations”

• Lns 244-48: Please indicate where the bottle described in these lines was purchased to avoid any argument that the bottle tested was not a genuine bottle but a counterfeit. – The following was added “The bottle was purchased at Walmart in Maryland at the same time as the purchase of the replicate bottle.”

Hopefully, the authors will consider all the comments to maintain the quality of the manuscript for the readership of the PLOS community.

As a reviewer, I am still sticking to my previous recommendation. Additional data is necessary to consider this paper as a full research manuscript, and need to improve and thoroughly discuss the tables and figures.

2. Line 21-23, Please state the quantitative value/data (mean±sd) for antioxidant

activity, phenolic concentrations, and flavonoid concentrations.

– Included one the range of CVs as an example, but it would be impractical to include the quantitative data for all of the supplements in a meaningful way within the abstract of the manuscript. In this case the CV is ideal as it gives an idea of variability that is standardized across values that are different by orders of magnitude.

My request is to include the mean±sd of antioxidant activity, phenolic concentrations, and flavonoid concentrations in the abstract section. The authors do not address this. – The coefficient of variation is calculated as a ratio of the mean to the standard deviation. It is specifically used in cases where the means are so different in a scale of magnitude that comparing one mean to the other would not be feasible. To include any kind of single overall mean ± std for antioxidants, phenolics and flavonoids would be useless. The values between supplements are so different that any sort of composite mean would not be representative of any of the values presented within the tables in the manuscript. Because the coefficient of variation is a measure of the relative standard deviation it is a much more efficient way to encapsulate what the reviewer wants in a very concise format that is also meaningful.

The description of statistical variation needs to be based on mean±sd, and the superscript should be stated for each treatment, which is based on the treatment variation. (This is also seen in the new document, line number 163 page 10). – I’m not sure how the reviewer expects for this to be included within the abstract since the values for each of these are so vastly different and that any composite mean would be meaningless. See comment above. 

10.Line 119. For antioxidant in vitro test, at least two assays needed; in addition, better also to look the Effective concentration (EC50). These make the antioxidant activity test/data to the standard and trustful.

– The reason for choosing only one type of assay for each type is included at the beginning of the discussion. “Recognizing that there are multiple methods of measuring antioxidants, phenolics and flavonoids it was decided that one type of assay for each should be conducted to more quickly determine if there

is justification for further investigation into the purity and consistency of the

supplements.”

Phenol and flavonoids are phytochemicals, which are considered as antioxidant content. My request is for antioxidant activity, like FRAP. In line with this calculating EC50 increases the readability of the manuscript, and also makes the data trustful. – The samples and supplements that we tested have long since passed the time where an additional antioxidant test, such as the EC50 could be performed on these bottles. However, in addition to the statement included in the first revision this statement has also been added to the discussion to indicate that further testing is justified and needed – “and that further testing involving different assays for antioxidant capacity and specific phenolics and flavonoids is justified for these supplements”

13. Line 148,149, please state the alpha level for the degree of significance.

- Done

The authors said done for this comment; however, the alpha level for the degree of significance is not in place. – The alpha level for each test is listed in the section under statistical analysis which is copied here. The alpha level was also included in the footnotes of the tables where appropriate. “Statistical Analysis. For each bottle of each supplement, the means and standard error were calculated for the water, methanol, and nitric acid extractions. In addition, the coefficient of variation (CV) was calculated for the water and methanol extractions. Comparisons among suppliers and bottles from the same supplier were done using the general linear model in the FitModel platform of JMP with an � level of 0.05 [38]. A runs test was also conducted to test for randomness of rankings among bottles, suppliers and supplements using Minitab� Statistical Software with an � level of 0.05 [39]. A significant runs test means suppliers and/or supplements clump together. To determine if there had been a change in the average values for the antioxidant, phenolic and flavonoid assays after two years of supplement storage the data was analyzed using a one-tailed t-test with an � level of 0.05 and a test value of < -25% change using Minitab� Statistical Software [39]. 

14. Line 170. Table 2 needs a general modification. This table should be with the numerical value (mean±se for each supplement and parameters). The mean values should be also differentiated using superscript letters if significant differences are found.

– This data is now split between 2 tables. One based on the means and SE for each bottle of each supplement and supplier. The other is based on the overall means for each supplier of each supplement (now tables 2 and 3).

The superscript stated in Table 2 and 3 are not correct, and is not statistical accepted. Some of the values do not have also the superscripts. – I’m not really sure why the reviewer states that the superscripts are not correct. I have made the following modifications to hopefully address the reviewers concerns. In table 2 a ^ was placed next to values from supplements where there is only one supplier so that comparisons among suppliers is not possible. In addition, I have placed a superscript letter next to all values, even those that are not significantly different. In addition our data set with statistical analysis has been reorganized and updated in the open source data link to make it easier to see all of the analysis. https://doi.org/10.7910/DVN/TRLTI3

17.Table 3. Better to describe as mean±sd, please.

– The table has been changed to show the CV for all values, but we believe that the CV is valuable in evaluating supplements with a wide range of values. Therefore we are including an expanded CV table in addition to the means and standard errors with accompanying superscripts in two new tables. See 15 above.

To my knowledge, CV does not indicate a statistical variation between treatments. For a reputable journal like PLOS ONE, a description of treatments using a statistical variation is important for the readers, and maintain the quality of the journal. – The purpose of the coefficient of variation is to allow for a snapshot of how much variation there is around a mean without reference directly to the mean and it’s attendant units. It is also useful in examining variation in tests that do not have similar means as it is a measure of the relative standard deviation (RSD)

• 

18. Table 3. needs to have superscript letters to show the degree of significance. –Done.

This is not properly done. – The table has been modified to include superscript letters for all values, even those without significant differences between bottles. Initially we had not included superscripts for the insignificant values to reduce the amount of clutter within the table.

19. Table 3. An antioxidant is a broad term that does not described as a single

parameter. It needs to be specific. In your case, FRAP. Please know that a single in vitro parameter is not to the standard to support the antioxidant activity of your samples. Please refer to the comment give in Line 119. – See response to comment 11 above.

This is not addressed. – To address this FRAP is included in all table titles in association with antioxidant capacity, but within the body of the tables we use only ‘A’ to symbolize this category in keeping with the single letter symbols ‘P’ and ‘F’ for phenolics and flavonoids.

20.Line 209 Table 4, please modify the title of the Table, state mean±sd with significant superscript letters. The range value can be indicated in your discussion.

– The title has been modified.

The title is modified; however, it is necessary to address the other requests. – Table 4 in the old manuscript is now table 6 in revision 1. For the metals we were simply testing for the presence versus absence of the metals within the supplements and did not conduct an ANOVA on the data.

---

## [Editor Report · Decision Letter 2]

22 Jan 2021

PONE-D-20-06630R2

Screening for consistency and contamination within and between bottles of 29 herbal supplements.

PLOS ONE

Dear Dr. Veatch-Blohm,

Thank you for submitting your manuscript to PLOS ONE. After careful consideration, we feel that it has merit but does not fully meet PLOS ONE’s publication criteria as it currently stands. Therefore, we invite you to submit a revised version of the manuscript that addresses the following points identified by the journal office:

1. Supplement stability tests were in some cases performed after the expiration date listed, but in others before. Please include some text discussing this as a limitation, and consider whether you could include subgroup analysis separating out the supplements that had expired from those that had not in the stability tests.

2. Please clarify whether the supplements were stored according to the package instructions, and were they unopened before testing? Storage conditions are currently described as having “mimicked storage conditions in a home”, but there is no formal description of these conditions (in terms of temperature, humidity etc.)

3. Please describe where all of the tested supplements were purchased e.g. online or a local store (please provide names).

4. Regarding the recalls (lines 83-88), in reading the archived FDA alerts, it seems that the companies themselves voluntarily recalled the products, not the FDA. We would recommend removing this sentence and the next in their entirety – they are not necessary to support the claims in this paper.

We look forward to receiving your revised manuscript.

Kind regards,

Jamie Males

Senior Editor

PLOS ONE

on behalf of

Catherine Nkirote Kunyanga, Ph.D.

Academic Editor

PLOS ONE
---

## [Author Response · Author response to Decision Letter 2]

14 Feb 2021

Changes have been made to the manuscript according to reviewer comments. A point by point response to the comments is included in the uploaded files

---

## [Decision Letter · Decision Letter 3]

4 Oct 2021

PONE-D-20-06630R3Screening for consistency and contamination within and between bottles of 29 herbal supplements.PLOS ONE

Dear Dr. Veatch-Blohm,

Thank you for submitting your manuscript to PLOS ONE. After careful consideration, we feel that it has merit but does not fully meet PLOS ONE’s publication criteria as it currently stands. Therefore, we invite you to submit a revised version of the manuscript that addresses the points raised during the review process.

We look forward to receiving your revised manuscript.

Kind regards,

Hikmet Aydin, MD, FAACC

Academic Editor

PLOS ONE

Journal Requirements:

Reviewers' comments:

Reviewer's Responses to Questions

**Comments to the Author**

1. If the authors have adequately addressed your comments raised in a previous round of review and you feel that this manuscript is now acceptable for publication, you may indicate that here to bypass the “Comments to the Author” section, enter your conflict of interest statement in the “Confidential to Editor” section, and submit your "Accept" recommendation.

Reviewer #2: All comments have been addressed

Reviewer #3: All comments have been addressed

Reviewer #4: (No Response)

Reviewer #5: (No Response)

2. Is the manuscript technically sound, and do the data support the conclusions?

Reviewer #2: Yes

Reviewer #3: Partly

Reviewer #4: Yes

Reviewer #5: Yes

3. Has the statistical analysis been performed appropriately and rigorously? 

Reviewer #2: Yes

Reviewer #3: Yes

Reviewer #4: Yes

Reviewer #5: Yes

4. Have the authors made all data underlying the findings in their manuscript fully available?

Reviewer #2: Yes

Reviewer #3: Yes

Reviewer #4: No

Reviewer #5: Yes

5. Is the manuscript presented in an intelligible fashion and written in standard English?

Reviewer #2: Yes

Reviewer #3: Yes

Reviewer #4: Yes

Reviewer #5: Yes

6. Review Comments to the Author

Reviewer #2: The authors have addressed the comments appropriately and I recommend the manuscript be accepted for publication.

Reviewer #3: 1) line 75: cancel comma before the references

2)Introduction: the cited references are a little bit old. Could you find newer literature?

3)Tables 3 and 6: what do you mean "0" is the same as not determined?

4)in Tables with results SD is very high in some cases.

5) Please, rewrite the conclusion.

6) Lack of novelty.

7) Lack of the validation of the methods.

Reviewer #4: Did the extraction protocol performed according to the total weight of the supplement, or according to weight of extract per gram in a supplement? No information is provided in the methods section in terms of the extract amount per weight.

Eventhough, morphological identification of fungal species is important and routinely used, for fungal systematics, taxonomic classification may not always perform well for species classifications. This would be a limitation which should be further addressed.

Similarly, the gross physical contamination of the samples were screened under a light-dissecting microscope; while no information regarding the standards for the methodology (such as AOAC International, American Spice Trade Association, pharmacopeia’s, FDA’s Macroanalytical Procedures Manual etc) is provided. Also comprehensive investigation are also based on chemical techniques along with Fourier transform-infrared (FTIR) analysis; which is also a limitation and is expected to be addressed.

Legal aspects in terms of chemical and microbiological quality of herbal supplements should be discussed within the respective directives.

Reviewer #5: The required revisions are made on the manuscript. It is really very important to search for the quality of herbal supplements and share the gathered results with public.

7. PLOS authors have the option to publish the peer review history of their article (what does this mean?). If published, this will include your full peer review and any attached files.

Reviewer #2: No

Reviewer #3: No

Reviewer #4: No

Reviewer #5: No

---

## [Author Response · Author response to Decision Letter 3]

3 Nov 2021

6. Review Comments to the Author

Reviewer #2: The authors have addressed the comments appropriately and I recommend the manuscript be accepted for publication.

Reviewer #3: 1) line 75: cancel comma before the references – Done

2)Introduction: the cited references are a little bit old. Could you find newer literature? – Four additional references from 2019 through 2021 were added to the introduction and one reference from 2020 was added to the discussion. The rest of the references we wanted to keep as they contribute to the justification for the importance of the research conducted. When the paper was first submitted almost 2 years ago at least some of the references were more recent.

3)Tables 3 and 6: what do you mean "0" is the same as not determined? – It is unclear to me what the reviewer is asking here. It is never stated in the paper that 0 is the same as not determined. When zero is listed as the mean of the value for the supplement it indicates that none of the pills tested of that supplement had any detectable amount of either the antixodant activity, phenolic or flavonoids or the metals in question. 

4)in Tables with results SD is very high in some cases. – This is addressed briefly in the discussion with the following statement -- (as demonstrated by the large standard errors)

5) Please, rewrite the conclusion. - done

6) Lack of novelty.

7) Lack of the validation of the methods.

Reviewer #4: 

1. Did the extraction protocol performed according to the total weight of the supplement, or according to weight of extract per gram in a supplement? No information is provided in the methods section in terms of the extract amount per weight.

- Added to the method sections- ‘All assays values were calculated and standardized based on the weight of the sample that was extracted.’

2. Even though, morphological identification of fungal species is important and routinely used, for fungal systematics, taxonomic classification may not always perform well for species classifications. This would be a limitation which should be further addressed. – The following statement was added to the discussion – “Morphological identification of the species indicates that there were potential species of concern, which indicate targets for further research to determine if the species tentatively identified are indeed problematic.” 

3. Similarly, the gross physical contamination of the samples were screened under a light-dissecting microscope; while no information regarding the standards for the methodology (such as AOAC International, American Spice Trade Association, pharmacopeia’s, FDA’s Macroanalytical Procedures Manual etc) is provided. 

- The following line was added to the methods. “The samples were scanned quickly for uniformity within a capsule and among capsules within the same bottle and between different bottles.”

- Within the results the following statement was changed “lacked gross physical contamination” and replaced with “were uniform in texture and color”

4. Also comprehensive investigation are also based on chemical techniques along with Fourier transform-infrared (FTIR) analysis; which is also a limitation and is expected to be addressed. – Included the following statement in the discussion - “An important next step is testing the supplements for the presence of and concentration of the active ingredient if present.” 

5. Legal aspects in terms of chemical and microbiological quality of herbal supplements should be discussed within the respective directives.

- It wasn’t exactly clear to me what the reviewer meant by this comment. I had it checked by the university’s research counsel and they also were not certain. In order to try to address it I added the following to the conclusion –‘The data presented here strengthens the case that lack of regulation leads to products that are not standardized and have the potential to either be ineffective or to cause harm. Currently in the United States manufacturers are to ensure safety of their products, but the FDA can only legally act to remove supplements when there is proof of harm, false statements or ineffective labelling [57].’

Reviewer #5: The required revisions are made on the manuscript. It is really very important to search for the quality of herbal supplements and share the gathered results with public.

7. PLOS authors have the option to publish the peer review history of their article (what does this mean?). If published, this will include your full peer review and any attached files.

Do you want your identity to be public for this peer review? For information about this choice, including consent withdrawal, please see our Privacy Policy.

Reviewer #2: No

Reviewer #3: No

Reviewer #4: No

Reviewer #5: No

---

## [Editor Report · Decision Letter 4]

11 Nov 2021

Screening for consistency and contamination within and between bottles of 29 herbal supplements.

PONE-D-20-06630R4

Dear Dr. Veatch-Blohm,

We’re pleased to inform you that your manuscript has been judged scientifically suitable for publication and will be formally accepted for publication once it meets all outstanding technical requirements.

Kind regards,

Hikmet Aydin, MD, FAACC

Academic Editor

PLOS ONE

---

## [Editor Report · Acceptance letter]

15 Nov 2021

PONE-D-20-06630R4 

Screening for consistency and contamination within and between bottles of 29 herbal supplements. 

Dear Dr. Veatch-Blohm:

I'm pleased to inform you that your manuscript has been deemed suitable for publication in PLOS ONE. Congratulations! Your manuscript is now with our production department. 

Kind regards, 

on behalf of

Dr. Hikmet Aydin 

Academic Editor

PLOS ONE